# Specialized coding patterns among dorsomedial prefrontal neuronal ensembles predict conditioned reward seeking

Roger I Grant[1], Elizabeth M Doncheck[1], Kelsey M Vollmer[1], Kion T Winston[1], Elizaveta V Romanova[1], Preston N Siegler[1], Heather Holman[1], Christopher W Bowen[1], James M Otis[1,2]*

[1]Department of Neuroscience, Medical University of South Carolina, Charleston, United States; [2]Hollings Cancer Center, Medical University of South Carolina, Charleston, United States

**Abstract** Non-overlapping cell populations within dorsomedial prefrontal cortex (dmPFC), defined by gene expression or projection target, control dissociable aspects of reward seeking through unique activity patterns. However, even within these defined cell populations, considerable cell-to-cell variability is found, suggesting that greater resolution is needed to understand information processing in dmPFC. Here, we use two-photon calcium imaging in awake, behaving mice to monitor the activity of dmPFC excitatory neurons throughout Pavlovian reward conditioning. We characterize five unique neuronal ensembles that each encodes specialized information related to a sucrose reward, reward-predictive cues, and behavioral responses to those cues. The ensembles differentially emerge across daily training sessions – and stabilize after learning – in a manner that improves the predictive validity of dmPFC activity dynamics for deciphering variables related to behavioral conditioning. Our results characterize the complex dmPFC neuronal ensemble dynamics that stably predict reward availability and initiation of conditioned reward seeking following cue-reward learning.

*For correspondence: otis@musc.edu

Competing interests: The authors declare that no competing interests exist.

## Introduction

The dorsomedial prefrontal cortex (dmPFC) has garnered considerable interest due to its dysregulation in diseases associated with disordered reward processing (*Chen et al., 2018*; *Courchesne et al., 2011*; *Dienel and Lewis, 2019*; *Holmes et al., 2018*; *Koob and Volkow, 2010*; *Ye et al., 2012*). These abnormalities include aberrant cell morphology and regional mass (*Courchesne et al., 2011*), abnormal activity patterns (*Dienel and Lewis, 2019*), and reduced behavioral performance on tasks that involve dmPFC activity (*Goldstein and Volkow, 2011*). Despite this knowledge, how unique cell types in dmPFC encode complex reward-related information to guide behavioral output is unclear, limiting our understanding of how reward processing occurs in healthy individuals as compared to those with neuropsychiatric diseases.

Neuronal activity in dmPFC neurons is observed to be heterogeneous in a variety of behavioral tasks (*Kim et al., 2016*; *Kobayashi et al., 2006*; *Matsumoto et al., 2003*; *Powell and Redish, 2014*), including those that involve reward-seeking behaviors (*Sun et al., 2011*; *Moorman and Aston-Jones, 2015*; *Otis et al., 2017*; *Otis et al., 2019*; *Sparta et al., 2014*). Recent studies have aimed to resolve this heterogeneity through cell-type specific recording strategies, such as in vivo calcium imaging in genetically or projection-defined neurons (*Otis et al., 2017*; *Otis et al., 2019*; *Siciliano et al., 2019*; *Ye et al., 2016*). Although some variability can be explained by identified

neuronal subpopulations, a vast majority of response diversity remains unexplained (*Otis et al., 2017*). For example, we recently demonstrated that many dmPFC excitatory neurons that project to the nucleus accumbens (NAc) show diverse responses to reward-predictive cues, with about two-thirds of the responding cells being excitatory responders and the other third being inhibitory responders. Similarly, dmPFC neurons that project to the paraventricular thalamus (PVT) also show diverse responses to reward-predictive cues, with about two-thirds of responding cells being inhibitory responders and the other one-third being excitatory responders (*Otis et al., 2017*; *Otis et al., 2019*). Finally, responses can be further subdivided by their temporal relation to the cue, as well as the reward (i.e., anticipation vs. consumption). Overall, although subpopulations of dmPFC output neurons could be labeled as 'generally excited' (e.g., dmPFC→NAc) or 'generally inhibited' (e.g., dmPFC→PVT), such assignment ignores much of the variability that is likely critical for behavioral control. Thus, a more thorough and unbiased means of defining the heterogeneous activity patterns among dmPFC output neurons is needed to understand how these neurons are engaged during reward-related behavioral tasks.

Here, we use in vivo two-photon calcium imaging to measure and longitudinally track the activity dynamics of single dmPFC excitatory output neurons throughout a Pavlovian conditioned licking task. We observe five unique neuronal ensembles after task acquisition that encode specialized information related to the sucrose reward, reward-predictive cues, and behavioral responses to those cues. These five ensembles differentially emerge across days during learning in a manner that improves the predictive validity of dmPFC population dynamics for deciphering reward delivery, cue presentation, and behavior. These adaptations were specific to a reward-predictive cue, but not another neutral cue, suggesting that the identified neuronal activity patterns are likely related to associative learning. Finally, we find that single neurons within dmPFC neuronal ensembles display both trial-to-trial (within sessions) and day-to-day (between sessions) stability after learning. Overall, our data reveal that heterogeneous excitatory neuronal ensembles in dmPFC evolve specialized coding patterns across cue-reward learning that are stably maintained after learning. Our results highlight the importance of ensemble-specific recording and manipulation strategies for understanding the function of dmPFC activity for reward processing.

## Results

### Unique excitatory neuronal ensembles in dmPFC during reward seeking

We employed a Pavlovian conditioned licking task wherein head-restrained mice were trained to associate one tone conditioned stimulus (CS+), but not another (CS-), with the delivery of a liquid sucrose reward (*Figure 1A–B*). Mice readily acquired this task across sessions (one session per day; see *Figure 1—figure supplement 1*), showing conditioned licking behavior between the CS+ offset and reward delivery (trace interval), but not the CS- offset and equivalent no-reward epoch during sessions on later days (deemed 'late in learning'; *Figure 1C–D*). A two-way ANOVA revealed a significant cue by session interaction for conditioned licking behavior (Δ lick rate; $F_{1,47}$ = 165.1; p-value < 0.001), and post hoc tests confirmed that mice licked significantly more during CS+ trials during late in learning sessions as compared with CS- trials during both sessions (p-values < 0.001) and CS+ trials during early in learning sessions (p-value < 0.001). Thus, conditioned licking behavior for the CS+, but not CS-, developed across days of training revealing that the cue-reward association had been acquired by sessions identified as 'late in learning'.

We monitored the activity dynamics of putative dmPFC excitatory output neurons during both 'early in learning' and subsequent 'late in learning' behavioral sessions using two-photon calcium imaging (via AAVdj-CaMK2α-GCaMP6s) in vivo (*Figure 1E–F*; *Figure 1—figure supplement 2*; *Figure 1—video 1*). The number of sessions early and late in learning was unique to each animal based on speed of acquisition in the task (see *Figure 1—figure supplement 1*) as well as the number of visualizable imaging planes (see *Figure 1F*). A single imaging plane (field of view [FOV]) was selected during each day of training, resulting in 28 FOVs recorded early in learning (n = 10 mice, 28 FOVs, 2092 neurons) and 21 FOVs late in learning (n = 7 mice, 21 FOVs, 1511 neurons; three mice did not reach late in learning due to headcap issues). Some, but not all, of these FOVs were visualized during both early and late in learning behavioral sessions (to allow cell tracking in Figure 4). Like previous reports (*Otis et al., 2017*; *Otis et al., 2019*), we found that neurons display excitatory and

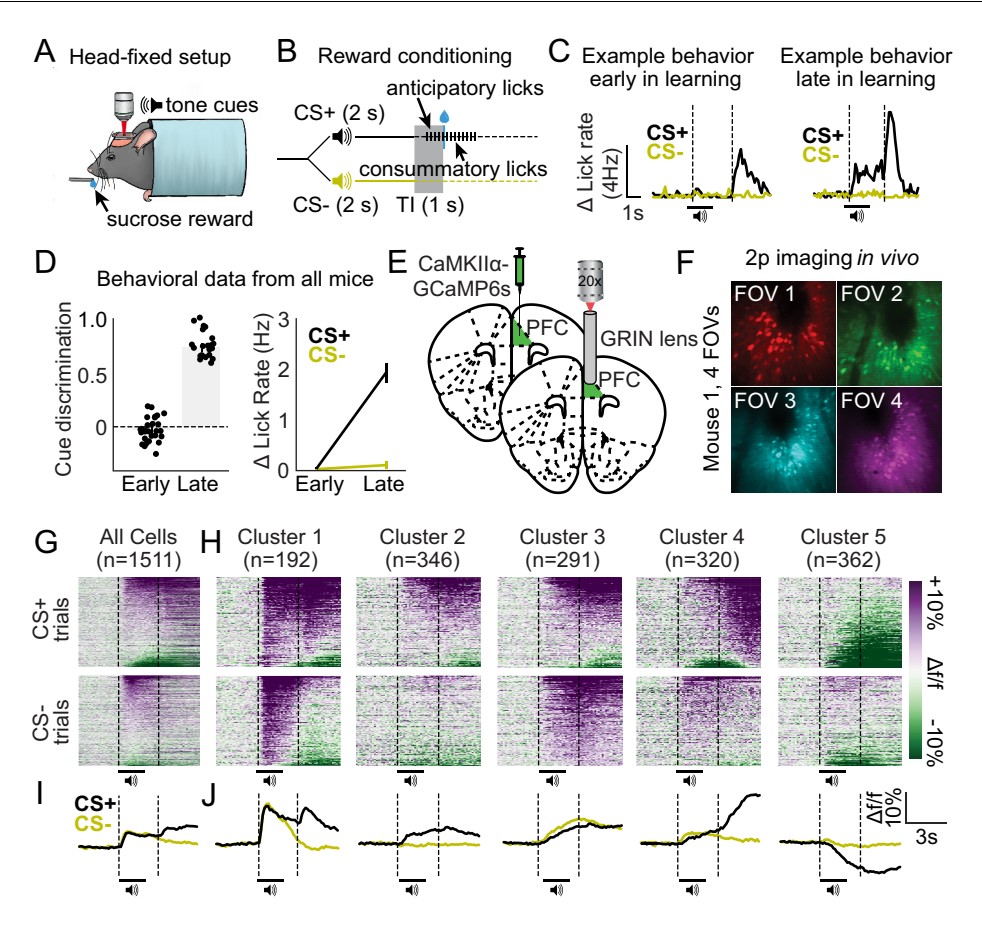

**Figure 1.** Distinct excitatory neuronal ensembles revealed in dorsomedial prefrontal cortex (dmPFC) during head-fixed Pavlovian conditioning. (**A**) Illustration of head fixation for reward conditioning experiments with concurrent two-photon imaging. (**B**) Schematic for reward conditioning experiments, in which the CS+ and CS- are presented in a random order 50 times each. The CS+ denotes the availability of a liquid sucrose reward following a 1 s trace interval (TI). Anticipatory licks are seen during the trace interval in well-trained mice. (**C**) Example behavior data during early in learning (left) vs. late in learning (right) behavioral sessions. (**D**) Cue discrimination scores (auROC; CS+ vs. CS-) and change in lick rate for each cue during early and late in learning behavioral sessions. Data presented as mean ± SEM. (**E–F**) Surgery schematic (**E**) allowing for in vivo imaging of GCaMP6s-expressing neurons (**F**). (**G–H**) Heat maps displaying the responses of all dmPFC neurons (**G**) and responses separated by cluster (**H**) aligned to the cues. (**I–J**) Line plots displaying the mean activity traces of all cells (**I**) and mean activity of all cells separated by cluster (**J**).

The online version of this article includes the following video and figure supplement(s) for figure 1:

**Figure supplement 1.** Behavioral task acquisition across days for each mouse.

**Figure supplement 2.** Visualizing dorsomedial prefrontal cortex (dmPFC) excitatory neurons within unique fields of view (FOVs) in a single mouse.

**Figure supplement 3.** Principal components analysis (PCA)-based clustering reveals unique silhouette scores for different clustering models.

**Figure supplement 4.** Similar neuronal ensembles defined by agglomerative and K-means clustering as compared with spectral clustering (see *Figure 1*).

**Figure supplement 5.** Estimated relative locations of each neuron across the anterior-posterior (A–P), medial-lateral (M–L), and dorsal-ventral (D–V) axes of dorsomedial prefrontal cortex (dmPFC).

**Figure 1—video 1.** Expression of GCaMP6s in dorsomedial prefrontal cortex (dmPFC) visualized via two-photon microscopy in vivo.

https://elifesciences.org/articles/65764#fig1video1

inhibitory responses to each cue and/or reward during late in learning behavioral sessions (*Figure 1G and I*). To better classify these post-learning activity patterns, we used a spectral clustering algorithm (*Namboodiri et al., 2019*) to isolate unique responses between recorded neurons (based on optimal clustering performance as compared to agglomerative and K-means clustering; see *Figure 1—figure supplements 3* and *4*). The analysis revealed the existence of five clusters or 'neuronal ensembles' that comprise most (but not all) of the response variability (*Figure 1H and J*) and are spatially intermixed within dmPFC (*Figure 1—figure supplement 5*). Each neuronal ensemble displayed a unique activity pattern during the CS+ and/or CS- trials, with qualitative analyses revealing the following dynamics: <u>Cluster 1</u>: excitatory responses during CS+, CS-, and reward delivery (n = 192 neurons from 5/7 mice; 17/21 FOVs), <u>Cluster 2</u>: excitatory responses during CS+ trials (n = 346 neurons from 7/7 mice; 19/21 FOVs), <u>Cluster 3</u>: excitatory responses during CS+ and CS- trials (n = 291 neurons from 7/7 mice; 20/21 FOVs), <u>Cluster 4</u>: excitatory responses during reward delivery (n = 320 neurons from 5/7 mice; 17/21 FOVs), and <u>Cluster 5</u>: inhibitory responses during CS + trials (n = 362 neurons; 7/7 mice; 21/21 FOVs). Overall, we find the existence of five unique ensembles among dmPFC excitatory output neurons, with each of these ensembles displaying unique activity patterns after learning in a Pavlovian conditioned licking task.

## Select excitatory neuronal ensembles in dmPFC predict behavioral performance during conditioned reward seeking

Considering that some neuronal ensembles were absent within some FOVs during a behavioral session, we determined whether the relative proportion of each neuronal ensemble in each FOV predicts behavioral performance. To this end, we quantified the proportion of neurons within each ensemble for all late in learning behavioral sessions (n = 21 sessions, 21 unique FOVs), and compared those values to cue discrimination licking scores (normalized auROC, CS+ vs. CS- lick rate). Overall, we find that proportion of neurons within Cluster 5 (which showed inhibitory responses during CS+ trials) predicts cue discrimination licking scores (*Figure 2A*). Pearson-R correlation values, found in the inset of each subpanel (*Figure 2A*), reveal a significant, positive relationship between cue discrimination licking scores and the percentage of neurons in Cluster 5 (p-value = 0.012). However, there was no significant correlation found for Clusters 1–4 (p-values > 0.2). These data are

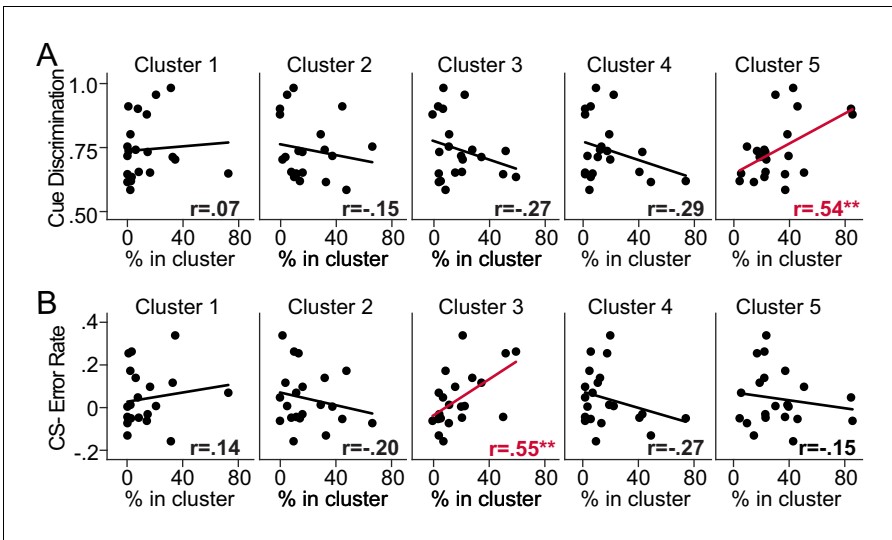

**Figure 2.** Behavioral performance is predicted by the relative percentage of dorsomedial prefrontal cortex (dmPFC) excitatory neurons within select ensembles. (**A**) Correlation plots separated by cluster displaying the relationship between cue discrimination behavioral scores after learning (auROCs: CS+ vs. CS- licking) and the percentage of detected neurons in a cluster during that behavioral session. The relative percentage of neurons in Cluster 5 positively predicted behavioral performance (\*\*p-value = 0.012). (**B**) Correlation plots separated by cluster displaying the relationship between CS- licking error rate after learning (auROC: CS- vs. baseline licking) and the percentage of neurons in a cluster during that behavioral session. The percentage of neurons in Cluster 3 positively predicted CS- licking error rate (\*\*p-value = 0.009).

particularly interesting considering previous findings showing that dmPFC→PVT neurons, which are also primarily inhibited during CS+ trials, are critical for the acquisition and expression of cue-induced reward seeking (*Otis et al., 2017*; *Otis et al., 2019*). These findings suggest that mice with greater numbers of neurons displaying inhibitory CS+ responses after learning, as in Cluster 5, may have improved behavioral performance in the appetitive learning task.

We also investigated whether the number of neurons within a particular neuronal ensemble predicted the probability that mice would increase licking during the CS- after learning (normalized auROC, CS- vs. baseline lick rate), which could be considered the initiation of a licking 'error'. Overall, we find that the relative proportion of neurons within Cluster 3 (which showed equivalent excitatory CS+ and CS- responses) predicts an increase in licking during CS- trials (*Figure 2B*). Pearson-R correlation values, found in each subpanel (*Figure 2B*), reveal a significant, positive relationship between the initiation of CS- licking 'errors' and the percentage of neurons in Cluster 3 (p-value = 0.009). However, there was no significant correlation found for other clusters (p-values > 0.2). These data suggest that mice with greater numbers of neurons that display equivalent, excitatory responses to both cues, as in Cluster 3, may be more likely to initiate reward seeking when rewards are not available.

## Excitatory neuronal ensembles in dmPFC display specialized coding during reward seeking

We find that dmPFC excitatory neuronal ensembles display unique activity patterns after developing Pavlovian conditioned behavioral responses, and that the relative proportion of neurons in each ensemble (specifically, Clusters 3 and 5) can predict behavioral task performance. Despite these findings, whether dmPFC activity patterns can be used to reliably infer environmental or behavioral events during the task is unknown. To this end, we trained a decoder to predict cue, reward, and licking events based on the activity dynamics of all neurons within each FOV (early in learning, n = 28 FOVs, 2092 neurons; late in learning, n = 21 FOVs, 1511 neurons). Overall, we find that dmPFC population dynamics within each FOV can be used to detect the presentation of the CS+, CS-, CS+ vs. CS- (cue discrimination), reward, and licking rate during late in learning sessions, whereas these activity patterns can be used to predict only the CS+, CS-, and CS+ vs. CS- (cue discrimination), but not reward delivery or licking rate during early in learning sessions (*Figure 3A*). ANOVAs, main effects of shuffling: CS+, $F_{1,92}$ = 37.29, p-value < 0.001; post hoc p-values < 0.05; CS-, $F_{1,92}$ = 22.88, p-value < 0.001; post hoc p-values < 0.05. CS+ vs. CS-, $F_{1,92}$ = 35.94, p-value < 0.001; post hoc p-values < 0.01; Reward, $F_{1,92}$ = 27.41, p-value < 0.001; early in learning post hoc p-value = 0.087, late in learning post hoc p-value < 0.001; Licking, $F_{1,92}$ = 23.94, p-value < 0.001; early in learning post hoc p-value = 0.199, late in learning post hoc p-value < 0.001. A heat map illustrating normalized decoding early and late in learning, and the change in that decoding across learning, reveals improved CS+ detection (post hoc p-value = 0.020), reward detection (post hoc p-value = 0.020), and lick rate prediction across learning (post hoc p-value = 0.009; other p-values > 0.37; *Figure 3B*). Overall, the activity dynamics of dmPFC excitatory output neurons can be used for cue detection, cue discrimination, reward detection, and prediction of licking after learning. However, activity in these neurons cannot be used to accurately infer reward delivery or licking during sessions early in learning, suggesting that the coding of these variables may be learning dependent.

The population dynamics of dmPFC excitatory output neurons can be used to predict environmental and behavioral factors related to conditioned reward seeking, but how unique dmPFC neuronal ensembles contribute to this information coding is unknown. Thus, we next trained a decoder to predict information related to the Pavlovian conditioned licking task based on the activity dynamics of neurons within each ensemble. Overall, we find superior decoding of the CS+, CS-, CS+ vs. CS- (cue discrimination), reward, and conditioned licking in select neuronal ensembles (*Figure 3C and D*). CS+: The timing of CS+ presentation could be decoded based on the activity of all cell clusters, although it was best predicted based on the activity of neurons within Cluster 1. CS-: The CS- was significantly predicted by Clusters 1, 3, 4, and 5, and was also best predicted based on the activity of Cluster 1. CS+ vs. CS-: Activity in Cluster 2, but not other clusters, could be used to significantly discriminate between the CS+ and CS-. Reward: Activity within Clusters 1, 4, and 5 could be used to detect the reward, with activity in Cluster 4 being the best predictor. Licking: Activity in Cluster 5, but not other clusters, could be used to decode conditioned licking. ANOVAs, main effects of shuffling: CS+, $F_{5,162}$ = 14.10, *p*-value < 0.001; post hoc p-values < 0.05 for all clusters; CS-, $F_{5,162}$ =

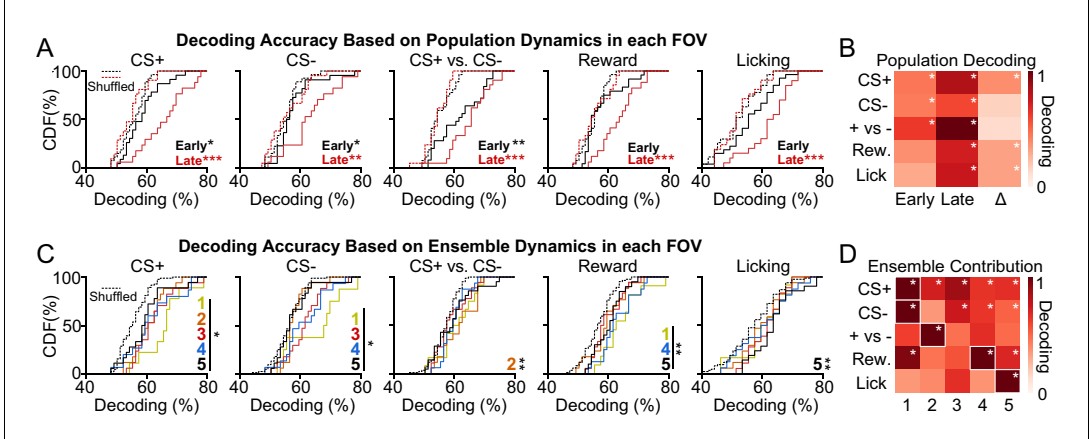

**Figure 3.** Activity of dorsomedial prefrontal cortex (dmPFC) excitatory neuronal ensembles can decode specialized information related to reward seeking. (**A**) Cumulative distribution frequency (CDF) plots illustrating the population decoding accuracy for variables related to conditioned reward seeking (CS+, CS-, CS+ vs. CS-, reward, and licking). Dotted lines refer to shuffled control data for early and late in learning. (**B**) Heat maps depicting population decoding accuracy early in learning (first column), late in learning (second column), and the change across learning (third column). Data have been normalized to CS+ vs. CS- late in learning to provide comparison of decoding strength across variables. (**C**) CDF plots illustrating the decoding accuracy of specific ensembles for variables related to conditioned reward seeking. Dotted lines refer to shuffled control data for all ensembles. (**D**) Heat maps depicting the contribution of each ensemble to decoding, with each column corresponding to a different ensemble. Data have been normalized to the maximum decoding strength by an ensemble for each variable to allow comparison of ensemble decoding strength across each variable. *p-value < 0.05; **p-value < 0.01; ***p-value < 0.001 for post hoc comparisons.

12.72, p-value < 0.001; post hoc p-values < 0.05 for Clusters 1, 3–5; post hoc p-value = 0.84 for Cluster 2. <u>CS+ vs. CS-</u>, $F_{5,162}$ = 3.46, p-value = 0.005; post hoc p-value = 0.004 for Cluster 2, post hoc p-values > 0.10 for other clusters; <u>Reward</u>, $F_{5,162}$ = 8.03, p-value < 0.001; post hoc p-values < 0.007 for Clusters 1, 4, and 5; post hoc p-values > 0.24 for Clusters 2 and 3. <u>Licking</u>, $F_{5,162}$ = 2.91, p-value = 0.015; post hoc p-value = 0.005 for Cluster 5, post hoc p-values > 0.16 for all other clusters. Overall, these data reveal that dmPFC excitatory neuronal ensembles predict select environmental and behavioral factors related to conditioned reward seeking after learning.

## Excitatory neuronal ensembles in dmPFC differentially develop during Pavlovian reward conditioning and are stable after learning

Two-photon microscopy enables visual tracking of single, virally labeled neurons across days (*Namboodiri et al., 2019*; *Otis et al., 2017*). Thus, we were able to track a subset of the above dmPFC excitatory output neurons from early to late in learning behavioral sessions (n = 5 mice, 9 FOVs, 416 neurons) to evaluate neuronal response evolution across learning (*Figure 4A*). Overall, we found that neurons in Cluster 1, which show excitatory responses to both cues and to the reward late in learning, also show robust responses to the same stimuli early in learning (*Figure 4B*). In contrast, neurons in Clusters 2–5 did not show obvious responses before learning during CS+ or CS- trials (*Figure 4B*), suggesting that their activity patterns evolved across conditioning and may therefore be reflective of learning.

To confirm that neurons in Clusters 2–5, but not Cluster 1, displayed CS+ and CS- response adaptations across learning, we compared responses from early in learning to late in learning behavioral sessions (*Figure 4C and D*). Two-way ANOVAs for Clusters 2–5 revealed an effect of behavioral session (F-values > 21.4; p-values < 0.001), whereas a two-way ANOVA for Cluster 1 did not reveal an effect of behavioral session (F-value = 0.034; p-value = 0.854). Post hoc comparisons confirmed no change in activity across sessions for Cluster 1 for CS+ or CS- trials (post hoc p-values > 0.20). In contrast, neurons in Cluster 2, 4, and 5 showed significant response adaptations for CS+ trials (post hoc p-values < 0.001), and Clusters 3 and 4 both showed a significant increase in response amplitudes across sessions for CS- trials (post hoc p-values < 0.02; all other p-values > 0.05 as shown in *Figure 4D*). Thus, neurons in Clusters 2–5 showed CS+ and/or CS- trial response adaptions from early in learning to late in learning behavioral sessions, whereas neurons in Cluster 1 did not.

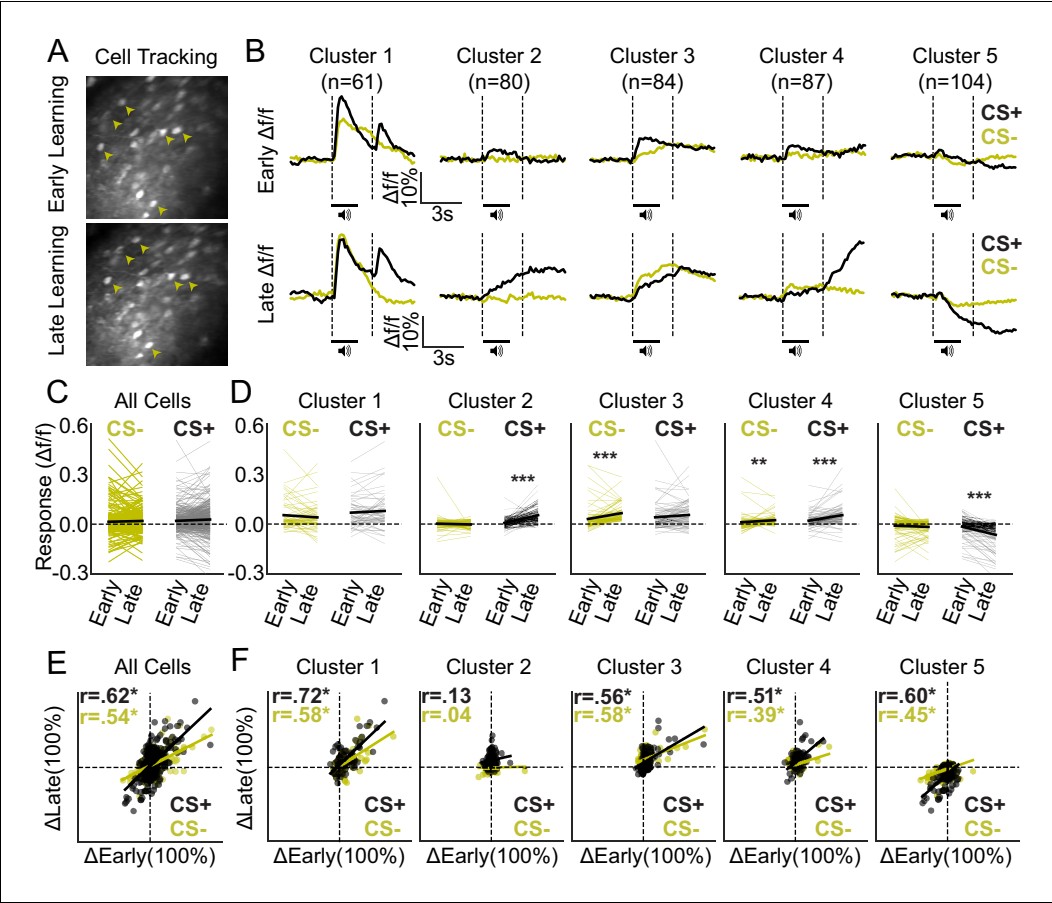

**Figure 4.** Ensemble-specific activity dynamics differentially evolve across learning. (**A**) Example field of view (FOV) showing the same neurons (arrowheads) from early (top) and late (bottom) in learning sessions, tracked across days. (**B**) Mean activity traces during CS+ and CS- trials for each ensemble early (top row) and late (bottom row) in learning. (**C**) Mean responses of all tracked neurons during CS+ and CS- trials early and late in learning. (**D**) Responses separated by cluster reveal adaptations in CS+ and/or CS- encoding for Clusters 2–5, but no significant changes for Cluster 1. ***p-value < 0.001; **p-values = 0.01. (**E**) Correlation plot displaying mean responses (baseline vs. cue/reward period) of all tracked neurons during CS+ and CS- trials early and late in learning. (**F**) Correlation plots separated by cluster displaying the mean response of each neuron early and late in learning. Pearson-R values are displayed in the top left corner for all cells and for each ensemble (**E–F**). *p-value < 0.05. The online version of this article includes the following figure supplement(s) for figure 4:

**Figure supplement 1.** Correlations in activity from early to late in learning are not due to similarities in activity across neurons within each field of view (FOV).

**Figure supplement 2.** Average neuron-to-neuron lags reveal signal coherence between neurons of the same cluster both early and late in learning.

Interestingly, responses during CS+ and CS- trials late in learning were highly predicted by responses early in learning for all neurons in combination (*Figure 4E*) and for individual cell clusters except Cluster 2 (*Figure 4F*). Pearson-R correlation values can be found in the inset of each subpanel (*Figure 4E and F*) and reveal positive correlations during CS+ and/or CS- trials that are significant for Clusters 1, 3, 4, and 5 (*denotes p-value < 0.05). Thus, although responses in Clusters 2, 4, and 5 adapted from early to late in learning behavioral sessions, responses in these clusters (and Cluster 1) early in learning could be used to predict their subsequent responses late in learning. Next, we confirmed that these correlated activity patterns were not simply due to similarities in responses between cells within tracked FOVs. To do so, we shuffled the tracking IDs for each neuron within each FOV and repeated the Pearson-R correlation analysis. Shuffling abolished the correlated response patterns between early and late in learning behavioral sessions (*Figure 4—figure supplement 1*), confirming that correlated activity patterns from the unshuffled datasets represent

similarities in activity across learning for individual neurons – rather than correlated activity patterns within tracked FOVs in general. In further support of this idea, cross-correlation analysis reveals little lag between neurons in the same cluster, as compared to neurons across clusters, both during early and late in learning behavioral sessions (*Figure 4—figure supplement 2*).

We next evaluated changes in neuronal activity within sessions, rather than between sessions, by examining trial-by-trial cue response evolution for each cluster. To do so, we used data from tracked neurons (*Figure 4*) such that we could examine cluster-specific adaptations for both early in learning and late in learning datasets. Mice showed consistent behavioral responses across trials both early in learning and late in learning (*Figure 5A and D*; n = 5 mice). Furthermore, the activity of dmPFC excitatory neurons (n = 9 FOVs, 416 tracked neurons) was consistent across trials during these behavioral sessions (*Figure 5B and E*), leading to a strong correlation in activity from trials at the beginning of the session (CS- and CS+ trials 1–10) vs. the end of the session (CS- and CS+ trials 41–50) for all neurons (*Figure 5C and F*). Pearson-R correlation values can be found in the inset of each subpanel (*Figure 5C and F*) and reveal positive correlations during CS+ and/or CS- trials (*denotes p-value < 0.001). Overall, these data reveal within-session response stability (behavioral and neuronal) during early in learning and late in learning datasets. Whether the observed response stability within sessions is also maintained between sessions after learning, however, remains unclear.

We next tracked the activity dynamics of dmPFC excitatory neurons after learning to determine if the defined neuronal ensembles remain stable or adapt across days (n = 3 mice, 4 FOVs, 142 neurons). Mice showed equivalent behavioral responses during these two late in learning behavioral sessions (*Figure 6A*), as a repeated t-test revealed no change in cue discrimination scores across days ($t_3 = 0.217$, p-value = 0.835). Furthermore, neuronal responses during both CS+ and CS- trials were highly correlated across these two behavioral sessions (*Figure 6B and C*; CS+: Pearson-R = 0.75, p-value < 0.001; CS-: Pearson-R = 0.30, p-value < 0.001), unless tracking IDs were shuffled for each FOV (*Figure 6—figure supplement 1*). Heat maps for each cluster reveal these highly correlated response patterns during both CS+ trials (*Figure 6D*) and CS- trials (*Figure 6E*). Overall, these data suggest that dmPFC excitatory neuronal ensembles display day-to-day response stability after cue-reward learning.

## Discussion

Here, we characterize unique excitatory neuronal ensembles in dmPFC that differentially predict behavioral task performance and encode specialized information related to Pavlovian conditioning.

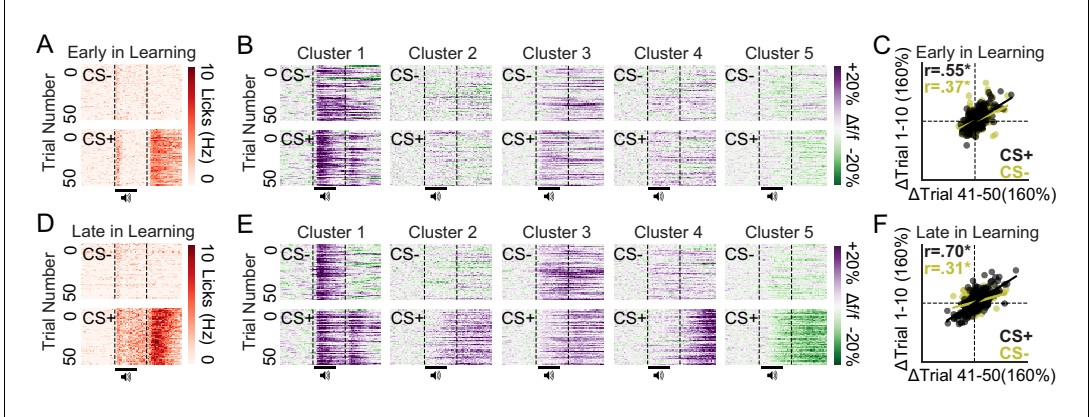

**Figure 5.** Stable conditioned licking and ensemble-specific cue responses within early and late in learning behavioral sessions. (**A**) Licking heat maps from sessions early in learning averaged across mice reveal stable licking across CS+ but not CS- trials, specifically during the 3 s reward epoch. (**B**) Activity heat maps from sessions early in learning averaged across neurons reveal consistent activity patterns across CS+ and CS- trials for each cluster. (**C**) Mean cue responses of all neurons in the first 10 trials during early in learning sessions were correlated with mean cue responses during the last 10 trials during early in learning sessions. (**E**) Licking heat maps from sessions late in learning averaged across mice reveal stable licking across CS+ but not CS- trials, specifically during CS+ delivery and subsequent 3 s reward epoch. (**F**) Activity heat maps from sessions late in learning averaged across neurons reveal consistent activity patterns across CS+ and CS- trials for each cluster. (**G**) Mean cue responses of all neurons in the first 10 trials during late in learning sessions were correlated with mean cue responses during the last 10 trials during late in learning sessions. *p-values < 0.001.

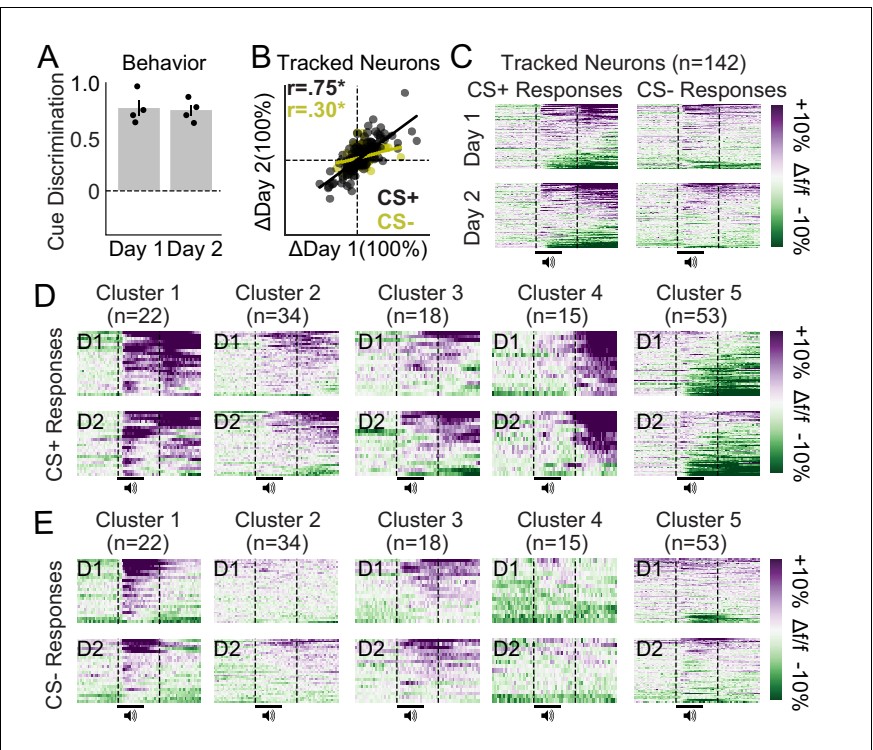

**Figure 6.** Ensemble-specific activity dynamics are maintained across days after learning. (A) Cue discrimination scores (auROC; CS+ vs. CS-) showing similar behavior across days after learning. Data presented as mean ± SEM. (B) Correlation plots displaying the change in activity during CS+ and CS- trials for all tracked neurons during two imaging sessions late in learning. These responses were highly correlated (*p-values < 0.001). (C) Activity heat maps for all tracked neurons separated by cue (columns) and day (rows) reveal similar population dynamics across days after learning. (D–E) Activity heat maps for each ensemble separated into CS+ trials (D) and CS- trials (E) confirm similar activity patterns across days after learning.

The online version of this article includes the following figure supplement(s) for figure 6:

**Figure supplement 1.** Correlations in activity across sessions after learning are not due to similarities in activity across neurons within each field of view (FOV).

The responses of each ensemble differentially emerge across learning in a manner that improves the predictive validity of dmPFC population dynamics for deciphering reward delivery, reward-predictive cue presentation, and task-related behavioral output. Considering the day-to-day stability of dmPFC activity dynamics after learning, our results suggest that each ensemble may be comprised of a unique set of cell types. Future studies that characterize the circuit connectivity, gene expression, and behavioral function of each neuronal ensemble, defined based on in vivo activity dynamics, are essential for understanding the dmPFC circuit contributions to reward processing.

Like previous studies, we find heterogenous activity patterns among dmPFC excitatory output neurons during reward seeking (*Murugan et al., 2017*; *Otis et al., 2017*; *Otis et al., 2019*; *Siciliano et al., 2019*) and use a spectral clustering algorithm to isolate five unique neuronal ensembles. Despite these findings, how each neuronal ensemble may be composed of unique cell types – for example, based on projection specificity – remains unclear. Previously, using the same Pavlovian reward-seeking task described here, we found that dmPFC→NAc neurons are 'generally excited' whereas dmPFC→PVT neurons are 'generally inhibited' following the presentation of a CS+, but not CS-, such that their overall activity patterns fit well within Cluster 2 (dmPFC→NAc) and Cluster 5 (dmPFC→PVT; *Otis et al., 2017*). Interestingly, in that study we found that optogenetic inhibition of dmPFC→PVT neurons facilitates cue-reward learning, whereas optogenetic activation of the pathway prevents learning and cue-evoked reward seeking. These data are consistent with the current findings showing that the proportion of cue-inhibited dmPFC neurons positively predicts behavioral performance in the task (see *Figure 2A*). Thus, we predict that Cluster 5 is accounted for in part by

dmPFC→PVT neurons, although further investigation is required to confirm this idea. Considering the heterogeneity found even in projection-specific recording studies in dmPFC (*Murugan et al., 2017*; *Otis et al., 2017*; *Otis et al., 2019*; *Siciliano et al., 2019*; *Vander Weele et al., 2018*), it is unlikely that a single projection pathway could be isolated to a single neuronal ensemble. To unravel the circuit connections of these unique cell types, it will therefore be necessary to selectively label neurons based on their in vivo activity dynamics, allowing for post hoc examination of their connectivity. Virally packaged fluorescent proteins that allow light-driven labeling of activated neurons, such as CaMPARI (*Fosque et al., 2015*), could be useful in this regard but also have limitations that would prevent precision labeling of selected neuronal ensembles (e.g., all activated cells would be labeled during UV light delivery, rather than only cells determined to be within a defined ensemble). Thus, development of novel technologies that allow robust labeling of experimenter-selected neurons are critical for identifying the projection profile of unique neuronal ensembles not only in dmPFC, but throughout the brain.

In addition to distinct circuit connectivity patterns, dmPFC neuronal ensembles may display differences in gene expression that could account for their unique activity dynamics. Although little is known about ensemble-specific gene expression in the dmPFC, cortical excitatory projection neurons are thought to express Camk2α (*Dittgen et al., 2004*), and there is evidence that the immediate early gene NPAS4 is upregulated in reward-responsive, but not aversion-responsive projection neurons (*Ye et al., 2016*). Additionally, layer-specific gene expression patterns may be present, such as in the case for genes encoding dopamine receptors (*Gaspar et al., 1995*), nicotinic acetylcholine receptors (*Verhoog et al., 2016*), noradrenergic receptors (*Santana et al., 2013*), and more (for review, see *Santana and Artigas, 2017*). However, recording experiments from subpopulations of dmPFC excitatory output neurons, defined based on gene expression, during reward seeking have been limited. Altogether, a more thorough characterization of ensemble-specific gene expression is needed to ascertain whether genetic differences account for unique activity patterns among dmPFC excitatory neuronal ensembles. Experiments involving single-cell sequencing ex vivo, such as patch-seq (*Cadwell et al., 2016*; *Cadwell et al., 2017*), could provide gene expression readouts from neuronal ensembles detected in vivo to improve our understanding of the gene expression differences that contribute to the evolution of distinct neuronal ensembles.

Our data showing learning-related, stimulus-specific activity patterns among dmPFC excitatory neuronal ensembles is consistent with previous studies. Previous investigations harnessing in vivo electrophysiology have found coordinated activity among undefined dmPFC cell populations during reward seeking, for example, related to consummatory behavior (licking) in a learning task (*Horst and Laubach, 2013*), initiation of complex behavioral strategies (*Powell and Redish, 2014*), recently committed errors in behavioral responding (*Powell and Redish, 2014*), and behavioral actions based on flexible information (such as rule shifting; *Bissonette and Roesch, 2015*; *Del Arco et al., 2017*; *Durstewitz et al., 2010*; *Powell and Redish, 2016*; *Rodgers and DeWeese, 2014*). Using waveform matching across days, one study even demonstrated day-to-day stability in behavioral strategy-related firing patterns (*Powell and Redish, 2014*), like another study showing stable *fos* expression, a marker of activated neurons, in dmPFC neurons across days in an appetitive conditioning task (*Brebner et al., 2020*). Altogether, these data are consistent with our findings showing day-to-day stability in variable-specific encoding patterns among dmPFC neuronal ensembles.

An important consideration for our study is the existence of heterogeneity not only between experimenter-defined cell clusters, but also within these clusters. We chose to use spectral clustering to define unique response dynamics in dmPFC neurons, as opposed to other methods (e.g., agglomerative, K-means), based on preliminary analyses (see *Figure 1—figure supplements 3* and *4*). Although these analyses suggested that spectral clustering provides the best separation of dmPFC neurons into groups, likely due to its ability to separate dynamic and non-singular response features, these methods overall are far from perfect. Simply put, we are trying to simplify a heterogeneous pattern of activity into homogeneous groups, which given current methodologies is only possible to some degree. Further advancement of clustering and other computational methodologies should continue to improve our ability to detect and understand unique response patterns within complex brain circuits.

One caveat to our study is the possibility that observed changes in neuronal activity could be driven by factors other than associative learning, such as non-associative learning (e.g., habituation) or feeding in general. However, there are several lines of evidence that suggest otherwise. First,

cluster-specific responses or adaptations in activity across learning were generally distinct for CS+ and CS- trials (see *Figure 4*), suggesting that the observed adaptations are related to cue-reward associative information. Second, cue discrimination, lick, and reward decoding improved over time, despite mice consuming the reward both before and after learning (see *Figure 5A and D*). Finally, previous evidence in the same behavioral task reveals that both excitatory and inhibitory responses among subpopulations of dmPFC pyramidal neurons are critical for cue-reward associative learning and cue-driven licking but not licking alone (*Otis et al., 2017*; *Otis et al., 2019*). Overall, evidence suggests that our identified adaptations in dmPFC activity dynamics are related to cue-reward associative learning. However, the possibility that there are components not related to associative learning should certainly be considered. Another consideration to note is that the observed conditioned licking responses could be influenced by the effects of water restriction, as well as the ratio of sucrose/water used as a reward in our behavioral paradigm (*Davey and Cleland, 1982*; *Harris and Thein, 2005*; *Tabbara et al., 2016*). However, we did not test the influence of these variables in the current experiments.

Here, we identify several distinct dmPFC excitatory neuronal ensembles during a Pavlovian conditioned licking task. Despite the apparent simplicity of this task, our findings reveal complex and specialized coding patterns among these heterogeneous neuronal ensembles, which are unlikely to be specific to one projection pathway or gene expression profile (*Otis et al., 2017*). Furthermore, our data suggest that unique aspects of reward seeking may be controlled by distinct neuronal ensembles. Functionally targeting each neuronal ensemble independently, such as through ensemble-specific single-cell optogenetic experiments, is therefore critical for understanding how these complex coding patterns control behavioral output (*Marshel et al., 2019*). Although our results improve our working knowledge of the unique excitatory neuronal ensembles within dmPFC during conditioned reward seeking, they also highlight critical gaps in the field of neuroscience that are important to resolve through new and emerging neurotechnologies.

## Materials and methods

### Subjects

Male and female C57BL/6J mice (8 weeks of age/25–35 g at study onset; Jackson Labs) were group-housed pre-operatively and single-housed post-operatively under a reversed 12:12 hr light cycle (lights off at 8:00 a.m.) with access to standard chow and water ad libitum. Experiments were performed in the dark phase and in accordance with the NIH Guide for the Care and Use of Laboratory Animals with approval from the Institutional Animal Care and Use Committee at the Medical University of South Carolina.

### Surgeries

Mice were anesthetized with isoflurane (0.8–1.5% in oxygen; 1 L/min) and placed within a stereotactic frame (Kopf Instruments) for cranial surgeries. Ophthalmic ointment (Akorn), topical anesthetic (2% Lidocaine; Akorn), analgesic (Ketorolac, 2 mg/kg, i.p.), and subcutaneous sterile saline (0.9% NaCl in water) treatments were given pre- and intra-operatively for health and pain management. Before lens implantation, a virus encoding the calcium indicator GCaMP6s (AAVdj-CaMK2α-GCaMP6s; UNC Vector Core) was unilaterally microinjected into the dmPFC (specifically targeting prelimbic cortex; 400 nL; anterior-posterior [AP], +1.85 mm; medial-lateral [ML], −0.50 mm; dorsal-ventral [DV], −2.45 mm). Next, a microendoscopic gradient refractive index lens (GRIN lens; 4 mm long, 1 mm diameter; Inscopix) was implanted dorsal to dmPFC (AP, +1.85 mm; ML, −0.50 mm; DV, −2.15 mm) as previously described (*Otis et al., 2017*; *Resendez et al., 2016*). A custom-made ring (stainless steel; 5 mm ID, 11 mm OD) was then adhered to the skull using dental cement and skull screws. Head rings were scored on the base using a drill for improved adherence. Following surgeries, mice received antibiotics (Cefazolin, 200 mg/kg, sc) and recovered with access to food and water ad libitum for at least 21 days. Histology was performed after the experiments to ensure virus placement in dmPFC and lens placement dorsal to dmPFC GCaMP6s-expressing neurons.

## Behavioral procedure

Mild water restriction facilitates appetitive learning in head-restrained mice, particularly when the reinforcer is sucrose mixed in water (*Guo et al., 2014*). Additionally, mild water restriction plus head-restraint results in minimal signs of distress while allowing simultaneous two-photon imaging across many trials of an appetitive behavioral task (*Guo et al., 2014*; *Goltstein et al., 2018*; *Otis et al., 2017*; *Otis et al., 2019*; *Namboodiri et al., 2019*). Thus, we used water restriction in combination with Pavlovian conditioning for a liquid sucrose reward to study appetitive learning in mice. Following recovery from surgery, mice were water restricted (water bottles removed from cages), and ~1 mL of water was delivered every day to a dish placed within each home cage (we gave less or more water during beginning phases of each experiment in attempt to calibrate weights to 90% of starting weights, which were 25–35 g). No health issues related to dehydration arose at any point during or after implementation of this protocol, as previously reported (*Guo et al., 2014*). Once mice reached 87.5–92.5% of their free drinking weight, they underwent 3 days of 30 min habituation sessions, during which they were head-restrained and received droplets of liquid sucrose (12.5% sucrose in water; ~2.0 µL per droplet, ~0.1 mL total per session) at random intervals through a gravity-driven, solenoid-controlled lick spout (see *Figure 1A*). Next, mice underwent head-fixed Pavlovian conditioning, wherein two conditioned stimuli (CS; 70 dB; 3 or 12 kHz as described in *Otis et al., 2017*) were randomly presented 50 times each (see *Figure 1B*). One tone (CS+) was paired to the delivery of a sucrose reward (12.5% sucrose in water, ~2.0 µL per droplet, ~0.1 mL total per session) after a 1 s trace interval, whereas the other tone (CS-) did not result in sucrose delivery. The trace interval was included to allow isolated detection of sensory cue- and sucrose reward-related neuronal activity patterns, as described previously (*Otis et al., 2017*). The inter-trial interval between the previous reward delivery (CS+ trials) or equivalent time epoch for unrewarded trials (CS- trials) and the next cue was chosen as a random sample from a uniform distribution ranging from 20 to 50 s. Cue discrimination was quantified using the normalized area under a receiver operating characteristic (2 × (auROC-0.5)) formed by the number of baseline-subtracted licks during the CS+ vs. CS- trace intervals (1 s epoch after tone offset). For all behavioral experiments, we classified sessions as 'early' or 'late' in learning based on animals' behavioral performance, quantified by their cue discrimination (early, any sessions before auROC < 0.3; late, any sessions after auROC > 0.31). Recordings from early in learning and late in learning sessions were on separate days, as only one session was given per day (see *Figure 1—figure supplement 1*). Mice received ~1 mL of water placed in a dish in their home cage after each conditioning session. Most mice readily acquired this task, but some required more sessions (without imaging) to discriminate between the cues. A small subset of animals had their head rings fall off during initial phases of the experiment, and thus their neural data is only included for early in learning behavioral sessions (n = 3 mice).

## Multiphoton imaging

We visualized and longitudinally tracked GCaMP6s-expressing dmPFC neurons throughout Pavlovian reward conditioning using a multiphoton microscope (Bruker Nano Inc) equipped with a hybrid scanning core with galvanometers and fast resonant scanners (>30 Hz; we recorded with four frame averaging to improve signal-to-noise ratio), GaAsP photodetectors with adjustable voltage and gain, a single green/red NDD filter cube, a long working distance 20× air objective designed for optical transmission at infrared wavelengths (Olympus, LCPLN20XIR, 0.45NA, 8.3 mm WD), a moveable objective in the X, Y, and Z dimensions, and a tunable InSight DeepSee laser (Spectra Physics, laser set to 920 nm, ~100 fs pulse width). Data were acquired using PrairieView software and converted into an hdf5 format for motion correction using SIMA (*Kaifosh et al., 2014*). FOVs were visualized from 0 to 300 µm beneath the GRIN lens and were selected before 'early in learning' imaging sessions. Each FOV was separated by at least 50 µm of objective movement in the Z-plane to avoid visualization of the same cells in multiple FOVs. Due to non-linear ray transformation introduced by the GRIN lens, this was especially important when imaging deeper FOVs. To ensure there was no signal bleed-through from superficial FOVs, a full cell layer was visualized between each FOV. Since neurons are ~20 µm in diameter, visualization of three layers (two imaging planes and an in-between plane) led to roughly 50 µm of z-movement and isolation of unique cell layers (*Figure 1—figure supplement 2*).

We attempted to image each FOV twice, once during a session early in learning and once during a session late in learning. The exact number of conditioning sessions per animal depended on the speed of learning and the number of visualizable FOVs in that animal. For example, if an animal had two FOVs, FOV 1 would be imaged on its first day of conditioning and FOV 2 would be imaged on its second day (these FOVs were 'early in learning' if cue discrimination scores remained below 0.3). Once the animal learned (cue discrimination scores above 0.31), both FOVs would be reimaged during separate conditioning sessions (these FOVs were 'late in learning'). Within each FOV, regions of interest around each cell were manually traced using the 'polygon selection' tool in FIJI (*Schindelin et al., 2012*). Care was taken to only assign regions of interest to visually distinct cells, and each region of interest was confirmed independently by an observer who was blind to the experimental conditions. In cases where neighboring cells or processes overlapped, regions of interest were drawn to exclude areas of overlap. The blind observer also ensured that cells were clearly in view (cells were not counted if the center-of-mass was not in focus) and that neurons were not visible in multiple FOVs (if they were, the imaging plane would not be used; n = 0). To do so, the blind observer examined z-stack videos to determine if the neurons between FOVs were independent (see *Figure 1—video 1*). Fluorescence trace extraction and all subsequent analyses were performed using custom-written Python code (*Namboodiri et al., 2019*; *Otis et al., 2019*).

## Data collection and statistics

Behavioral sessions were controlled through a custom MATLAB graphical user interface connected to Arduino and associated electronics. Transistor-transistor logic pulses between the Arduino and the microscope were used to start and stop imaging and behavioral programs, and to allow frame timestamp collection for post hoc synchronization of the behavioral and imaging data. Behavioral data were recorded and extracted using MATLAB, analyzed and graphed using Python and/or Graphpad Prism, and figures were produced using Adobe Illustrator. Behavioral data were presented as normalized auROC 'cue discrimination' scores (2 × (auROC-0.5)), comparing licking rates during the 1 s period between each CS+ and reward (1 s trace interval) or CS- and equivalent no-reward epoch (1 s trace interval). A cue discrimination score of −1.0 would therefore suggest more licking during all CS- trials vs. CS+ trials. In contrast, a score of +1 would suggest more licking on all CS+ trials vs. CS- trials. Following data collection, a two-way ANOVA was used to compare baseline-subtracted lick rates (Δ lick rate; calculated as: 1 s trace interval licking frequency – 3 s baseline licking frequency), followed by Bonferroni multiple comparisons tests if appropriate (*Otis et al., 2017*).

Fluorescence signals from each cell were extracted following motion correction using custom-written Python code. Activity in each cell was then aligned to each CS- and CS+ trial, which included a ~3 s baseline (23 frames), ~3 s cue period (including CS+ or CS- trace intervals; 23 frames), and ~3 s reward period (following sucrose delivery for CS+ trials or equivalent no-reward epoch during CS-trials; 23 frames). This resulted in 69 frames for each CS+ and CS- trial, which was then combined into a 138-column vector of data points for each cell, referred to as a peristimulus time histogram (PSTH). Due to the robust responses of dmPFC neurons late in learning (as also seen in *Otis et al., 2017*), data included within a 138 × 1511 vector (138 frames, 1511 neurons) from late in learning sessions then underwent principal components analysis to reduce their dimensionality in preparation for clustering, an unbiased means of identifying putative neuronal ensembles in dmPFC. We used an analysis and code that was previously created by others and kindly shared (full description of clustering in *Namboodiri et al., 2019*). The principal components were determined using the point of inflection on a scree plot, which graphs the PSTH variance explained vs. an increasing number of principal components (*Figure 1—figure supplement 3A*). Beyond this inflection point, minimal variability can be explained by additional principal components. The data were then projected onto the subspace formed by these principal components (*Figure 1—figure supplement 3B*), which was fed into the clustering algorithm. We used the Scikit-learn function *sklearn.cluster.SpectralClustering* to perform spectral clustering on these data, which uses a k-nearest neighbor connectivity matrix to create clusters. The optimal number of clusters and nearest neighbors were determined by checking a range of values for each and choosing the parameters with the maximum silhouette score. After spectral clustering, each neuron was assigned a label based on its corresponding cluster. Results from spectral clustering, including the silhouette scores and formed ensembles, were compared to agglomerative ('hierarchal') clustering and K-means clustering datasets, each of which was also performed using Scikit-learn. Silhouette plots (*Figure 1—figure supplement 3C–E*) revealed that

spectral clustering performed better than agglomerative and k-means clustering algorithms, although all three clustering algorithms resulted in separation of neurons in a manner that resulted in similar neuronal response dynamics across five clusters (see *Figure 1G–H* for spectral clustering and *Figure 1—figure supplement 4* for agglomerative and k-means clustering). Due to the improved performance of spectral clustering over agglomerative and k-means, we used spectral clustering for all additional datasets as previously described (*Namboodiri et al., 2019*).

Spatial mapping was performed for all FOVs recorded during late in learning behavioral sessions, since the clustering analysis was performed using activity patterns measured during those sessions (see above). To estimate the relative spatial locations along the AP and ML axes, we measured the central point of each neuron within a 512 × 512 standard deviation project image following XY plane transformation (to account for GRIN lens light reflection; *Figure 1—figure supplement 5A*). Additionally, we estimated the relative spatial locations along the DV axis by measuring the amount of objective movement along the Z axis after focusing on the bottom of the lens (*Figure 1—figure supplement 5B*). Kernel density estimation for cell locations relative to other cells are quantified in histograms along each axis in *Figure 5*. It should be noted that due to non-linear ray transformation through the GRIN lens, variability in lens angles, variability in head ring tilt, and general difficulty identifying the exact location of each neuron post-mortem, these anatomical locations are highly approximate and are not to a precise scale. Despite these limitations, the data provide a general idea of the spread of neurons, grouped by cluster, along the AP, ML, and DV axes in our experiments.

We compared the behavioral performance of each mouse with the number of neurons detected per ensemble in the corresponding FOV visualized during that behavioral session. Specifically, we used Pearson's correlations to compare cue discrimination scores (normalized auROC, CS+ vs. CS-) with the number of neurons detected in each ensemble during late in learning behavioral sessions (one unique FOV per session; *Figure 2A*). Additionally, we used Pearson's correlations to compare the initiation of licking 'errors', wherein mice increased licking rates following the presentation of the CS- (normalized auROC, CS- vs. baseline epoch), with the number of detected neurons per ensemble (*Figure 2B*).

Decoding analyses were employed to determine whether neuronal activity within each FOV, and each ensemble within each FOV, could predict variables within the behavioral paradigm better than by chance. Specifically, we used these neuronal data to inform a decoder to predict (1) <u>CS+</u>: 2 s CS+ epoch vs. 2 s baseline, (2) <u>CS-</u>: 2 s CS- epoch vs. 2 s baseline, (3) <u>CS+ vs. CS-</u>: 2 s CS+ epoch vs. 2 s CS- epoch, (4) <u>Reward</u>: 1 s epoch starting at sucrose delivery vs. 1 s pre-sucrose baseline, and (5) <u>Licking</u>: relative licking rate for each mouse during CS+ trials; 6 s epoch starting from CS+ onset to include anticipatory licking and sucrose consumption. Decoding was performed on the entire dataset (population decoding, *Figure 3A–B*), and separately for each ensemble (ensemble contribution, *Figure 3C–D*). To perform these analyses, we used a binary decoder as described previously (*Otis et al., 2017*; *Otis et al., 2019*), implemented using the Scikit-learn functions *sklearn.discriminant_analysis*, *sklearn.svm*, and *sklearn.decomposition*. For population decoding (decoding based on the activity of all neurons within each FOV), decoding scores were normalized to the maximum value observed for a decoded variable (which was CS+ vs. CS-, late in learning). For ensemble-specific decoding (decoding based on the activity of all neurons within each ensemble within each FOV), scores were normalized to the maximum for each variable, such that the contribution of each ensemble to variable decoding could be evaluated. To determine whether decoding performance was significantly better than chance, we compared the decoding accuracy to that of randomized 'shuffled' data using two-way ANOVAs (population decoding) or one-way ANOVAs (ensemble contribution), followed by program-recommended post hoc analysis (Dunnett's for one-way ANOVA, Tukey's for two-way ANOVA). Because shuffled data were not significantly different between ensembles, these data were combined to improve data visualization and simplify analysis (dotted lines in *Figure 3C*).

Specific neurons could be reliably identified across days based on structure and relative position within each FOV (*Figure 4A*), allowing us to evaluate the evolution and maintenance of activity in single neurons across days. To do so, single-cell tracking was performed from early to late in learning behavioral sessions to determine neuronal response evolution across learning (*Figure 4*). Additionally, tracking was performed across two late in learning behavioral sessions each separated by a minimum of 48 hr to evaluate post-learning response adaptation or maintenance (*Figure 6*). Cell

tracking was performed by a student blinded to experimental conditions to reduce the potential for experimenter-related biases. Cells were identified based on relative position and morphology across days, with conservative longitudinal tracking to prevent between-cell comparisons. Following initial cell tracking, a second experimenter confirmed accurate day-to-day cellular co-registration for cross validation. Two-way ANOVAs were used to compare the mean response (3 s baseline vs. 6 s cue/reward epoch) of each cluster across cues (CS+ vs. CS-) and time (e.g., early vs. late in learning; *Figure 4D*). Additionally, Pearson's correlation analyses were used to determine linear association of responses for each cluster across days (*Figures 4E–F* and *6B*) or across trials within sessions (*Figure 5C and F*). We performed control Pearson's correlation analyses for each tracking experiment wherein the student-identified tracking IDs within each FOV (but not between) were shuffled (*Figure 4—figure supplement 1*; *Figure 6—figure supplement 1*) to confirm that correlated activity patterns required accurate cell tracking. Finally, we used a cross-correlation analysis to identify lags for correlated activity patterns for each neuron as compared to all other neurons in each FOV for tracked datasets, which we then separated by cluster and averaged across each cluster-to-cluster comparison. To perform the cross-correlation analysis, we used the fluorescence signal of each neuron measured across the entire session to feed the SciPy functions *scipy.signal.correlation* and *scipy.signal.correlation_lags*. Optimal lags were then averaged across all neurons for each cluster-to-cluster comparison and plotted within a heat map (*Figure 4—figure supplement 2*). It should be noted that the cross-correlation analysis has limitations due to the temporal dynamics of the calcium indicator and general two-photon imaging speed, preventing spike-to-spike comparisons across neurons. Rather, the analysis allows assessment of general correlations, and associated lags, across neurons on a longer timescale (hundreds of milliseconds).

Raw behavioral data and fluorescent signals from imaging datasets, along with example images presented throughout this manuscript, can be found in an open-source data repository (https://doi.org/10.5061/dryad.xksn02vg8).

## Acknowledgements

The authors would like to thank Vijay MK Namboodiri and Garret D Stuber for creating and sharing clustering codes for imaging analysis.

## Additional information

### Funding

| Funder | Grant reference number | Author |
| --- | --- | --- |
| National Institute on Drug Abuse | R01-DA051650 | James M Otis |
| Medical University of South Carolina | | James M Otis |

The funders had no role in study design, data collection and interpretation, or the decision to submit the work for publication.

### Author contributions

Roger I Grant, Conceptualization, Formal analysis, Investigation, Methodology, Writing - original draft, Writing - review and editing; Elizabeth M Doncheck, Kelsey M Vollmer, Kion T Winston, Elizaveta V Romanova, Preston N Siegler, Heather Holman, Investigation; Christopher W Bowen, Conceptualization, Formal analysis, Supervision, Investigation, Writing - original draft, Writing - review and editing; James M Otis, Conceptualization, Formal analysis, Supervision, Investigation, Methodology, Writing - original draft, Writing - review and editing

### Author ORCIDs

Roger I Grant (iD) https://orcid.org/0000-0003-4609-5773
James M Otis (iD) https://orcid.org/0000-0003-0953-9283

## Ethics

Animal experimentation: Experiments were performed in the dark phase and in accordance with the NIH Guide for the Care and Use of Laboratory Animals with approval from the Institutional Animal Care and Use Committee at the Medical University of South Carolina (Approval ID: IACUC-2018-00363; Renewed November 30, 2020).

## Decision letter and Author response

Decision letter https://doi.org/10.7554/eLife.65764.sa1
Author response https://doi.org/10.7554/eLife.65764.sa2

# Additional files

## Supplementary files

- Transparent reporting form

## Data availability

All code and data generated for this study are available on Dryad Digital Repository, accessible here: https://doi.org/10.5061/dryad.xksn02vg8. Raw imaging videos are available on request.

The following dataset was generated:

| Author(s) | Year | Dataset title | Dataset URL | Database and Identifier |
|---|---|---|---|---|
| Grant RI, Doncheck EM, Vollmer KM, Winston KT, Romanova EV, Siegler PN, Holman H, Bowen CW, Otis JM | 2021 | Behavioral and Imaging data for: Grant et al. (2021). Specialized coding patterns among dorsomedial prefrontal neuronal ensembles predict conditioned reward seeking | http://dx.doi.org/10.5061/dryad.xksn02vg8 | Dryad Digital Repository, 10.5061/dryad.xksn02vg8 |

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
