## [Decision Letter]

**Acceptance summary:**

This study offers novel insight on neurophysiological mechanisms occurring in the prefrontal cortex (PFC) during the learning of cue-reward associations. The results will be of great interest to scientists in the behavioral and systems neuroscience fields, but also to the broader scientific audience due to the use of challenging in vivo techniques, computational analyses, and statistical methods.

**Decision letter after peer review:**

Thank you for submitting your article "Specialized coding patterns among dorsomedial prefrontal neuronal ensembles predict conditioned reward seeking" for consideration by *eLife*. Your article has been reviewed by 3 peer reviewers, and the evaluation has been overseen by a Reviewing Editor and Kate Wassum as the Senior Editor. The following individuals involved in review of your submission have agreed to reveal their identity: Maria M Diehl (Reviewer #2); Anthony Burgos-robles (Reviewer #3).

Essential Revisions:

The reviewers agree that the authors' identification of dmPFC neuronal ensembles with heterogeneous coding patterns offers important insight about the neurophysiological mechanisms governing cue-reward learning. However, as independently outlined below by each one of the reviewers, there are multiple aspects of the study that must be strengthened before the paper can be considered for publication in *eLife*. With the exception of the optogenetic behavioral manipulations requested by Reviewer # 3, we consider that all other concerns raised by the reviewers must be addressed in full. Specifically, the authors must address:

1) All technical concerns regarding the imaging technique that were raised by Reviewer # 1.

2) All statistical and data analysis concerns raised by Reviewers #1, 2 and 3.

3) Additional clarifications of methods and analyses, and an improved discussion as suggested by the reviewers.

4) The concerns about the behavioral design raised by Reviewer #1.

5) Due to the lack of causality, revise the text to soften the language a bit in some some of the sentences describing the interpretation.

Please notice that addressing these concerns will require, at a minimum, the incorporation of new data analyses, validation data, and an extensive revision of the manuscript's text.

If you have not already done so, please include a key resource table.

*Reviewer #1:*

The manuscript addresses a critical question in cortex and neuroscience in general – how do neuronal coding patterns lead to behavioral outputs and learned behaviors? While the manuscript takes a technically innovative approach there are multiple issues with the behavioral design, imaging, and interpretation.

There are issues with the using multiple imaging planes in each animal and with the longitudinal co-registration. Regarding the FOVs, the authors report that each imaging plane was separated by 50uM. However, since GRIN lenses display non-linear ray transformations in both the lateral and axial planes, movement of the external objective in 50uM steps cannot be assumed to produce a 50uM change in imaging plane. Even if we are to accept that no cells were double counted, a more critical issue is that collection efficiency, and therefore SNR of the recording, will be altered as a function of the distance of each neuron from the ideal focal plan of the implanted GRIN endoscope. Additionally, more information is needed on the longitudinal registration including how these data were validated and the percentage of neurons tracked.

With respect to the behavior, it is not clear whether the changes are the result of reward learning or more simply related to non-associative variables like habituation, lick rate, or volume consumed.

The manuscript from Grant et al., explores heterogeneity in coding patterns of mPFC pyramidal neurons during reward learning. The manuscript addresses a critical question in cortex and neuroscience in general – how do neuronal coding patterns lead to behavioral outputs and learned behaviors? While the manuscript takes a technically innovative approach there are multiple issues with the behavioral design, imaging, and interpretation. These issues are addressable, and the manuscript has potential for high impact in the field, but to support the current conclusions would require significant additional analysis and experimentation. Issues are listed below:

1. There are major issues with the imaging methodologies, particularly with the using multiple imaging planes in each animal and with the longitudinal co-registration. Regarding the FOVs, the authors report that each imaging plane was separated by 50uM. However, since GRIN lenses display non-linear ray transformations in both the lateral and axial planes, movement of the external objective in 50uM steps cannot be assumed to produce a 50uM change in imaging plane. Indeed, in the representative images the same vasculature can be seen in multiple planes, and the veins in this area often smaller than 50uM. Though it is difficult to discern, it appears that the same cell constellations appear in multiple planes in the representatives.

2. Even if we are to accept that no cells were double counted, a more critical issue for the current claims of the manuscript is that collection efficiency, and therefore SNR of the recording, will be altered as a function of the distance of each neuron from the ideal focal plan of the implanted GRIN endoscope. This is a potentially critical flaw without significant additional analysis. For example, all of the clustering analysis could be highly biased by the number of neurons that were included from each imaging plane which is likely to vary from animal to animal. While this is always somewhat of a concern with GRIN imaging, because 3-4 imaging planes were used in each animal and that clustering analysis was performed on pooled data it is possible that difference in SNR across imaging planes is driving many of the effects in the manuscript.

3. Regarding longitudinal registration, minimal methodological information is provided which is concerning given that this a notoriously difficult endeavor especially in dense recording such as these data. How were these data validated? Was the data set scored by a second experimenter for cross validation? What percent of neurons were tracked? Was any network analysis of cell location used to verify results?

4. While the issues with the imaging are critical to address, it is likely that in depth analysis could resolve the problems without the need for additional experiments. However, there is also some problems with the behavioral design – these will either require additional experiments or require that the claims of the manuscript be altered. All of the changes in mPFC dynamics that occur across the behavioral task are claimed to be related to reward learning, but there are several processes that are not parsed in the task design. For example, would some of these changes happen with habituation? Would some of these changes happen with lick rate, volume consumed? None of those are dependent on associative learning and could just as strongly predict the changes that are seen in the dataset.

5. What is the justification for using water restriction combined with a sucrose solution in water as a reinforcer? Given that sucrose functions as a reinforcer without water restriction and that water functions as reinforcer under water deprived conditions, it is unclear whether the water or the sucrose is functioning as the primary reinforcer.

*Reviewer #2:*

Grant et al., used two-photon calcium imaging of dorsomedial prefrontal cortical neurons to examine the neuronal ensemble activity during a sucrose conditioning task in head-fixed mice. Using a spectral clustering algorithm, the authors characterized ensemble activity into 5 distinct clusters whose activity correlated with various aspects of the task: CS+ responding, CS- responding, CS discrimination, reward responding, and licking behavior. Cluster 1 exhibited excitatory responses to both CSs and reward delivery, Cluster 2 exhibited excitatory responses to CS+ only, Cluster 3 exhibited excitatory responses to both CSs, Cluster 4 exhibited excitatory responses to reward delivery only, and Cluster 5 exhibited inhibitory responses to CS+. Next, the authors determined whether each Cluster predicted licking behavior across each conditioning session for each mouse. They found that the proportions of neurons in Cluster 5 positively correlated with successful licking behavior (licking in response to CS+), whereas proportions of neurons in Cluster 3 positively correlated with licking errors (licking in response to CS-). The authors were next interested in whether the neural activity across all dmPFC neurons (regardless of cluster ensembles) predicted task events or animal licking behavior during early vs. late in learning sessions of the task. Overall, CS presentation, CS discrimination, reward delivery, and licking rate were predicted by dmPFC activity during late, but not early, in learning. Taking into account the cluster ensembles, the authors also identified whether the activity of each cluster could predict CS presentation, CS discrimination, reward delivery, and licking rate. Finally, the authors assessed whether the cluster ensembles remain stable after learning; Cluster 1 showed robust responses to both CSs and reward delivery during both early and late in learning sessions of the task, whereas Clusters 2-5 did not, suggesting that these latter Clusters changed their activity patterns as a function of learning. The authors conclude that excitatory neuronal ensembles in dmPFC differentially predict events and behaviors during a sucrose conditioning task and the responses can change across learning. The conclusions of the paper are supported by the data; however, some aspects of the paper need to be clarified, additional analyses performed, and more interpretation of the data is warranted in order to strengthen the importance of this study.

More clarity is needed on how early and late session classification and whether the results would be similar if data were based on trial number instead of auROC across session.

The cluster analysis would benefit from the addition of location information to determine whether the 5 identified clusters are anatomically segregated. The data would also benefit from a cross-correlation analysis to reveal if there are any significant interactions between neurons within the same FOV, determine whether they are from the same cluster, whether cells active early in learning interact with cells that are active late in learning, and whether these cross-correlations remain stable after learning.

1. Behavioral sessions were classified as either "early in learning" or "late in learning" and are defined based on the animal's performance (using auROC) across multiple sessions of the sucrose conditioning task. The authors perform an independent t-test comparing performance in early vs. late in learning sessions; however, these 2 groups of sessions were pre-defined by the animal performance itself. Therefore, it is inappropriate to use a statistical test since the periods of early and late were selected based on the behavioral data (i.e. doing a statistical test on data that was pre-selected). A statistical test is not needed here, but the authors should emphasize that the number of early vs late in learning sessions were unique to each animal and selected based on their performance (this is only mentioned in the methods, but authors should consider reiterating this point in the results). As a reader, it was very difficult to understand how early and late in learning was defined and I had to go back and read the methods multiple times and look through the results for clarification. I initially thought early vs. late in learning referred to early trials vs. late trials within a session. If early and late was defined based on trial number instead of auROC across sessions, are the behavioral results similar to what was reported? On average, how many sessions did it take for each mouse to reach criteria?

2. Because I was confused about early vs late in learning being within session vs. across session, I was also confused about the data analyzed in Figure 4. How many sessions were in early vs late? Was the miniscope lens advanced after each conditioning session? Was there a subset or a different set of neurons that were recorded from across multiple sessions/days? Also, why do the traces in Figure 1I look very identical to the traces in Figure 4B (lower)? Are the cells analyzed in Figure 4 a subset of those shown in Figure 1? Please clarify. I think it would also be helpful to use the same terms consistently throughout the text when referring to the conditioning sessions and clearly state at the beginning of the results that one session was recorded per day and the lens was advanced to a new FOV (if that was the case) – sometimes sessions are referred to as FOVs or different days.

3. PFC-PVT neurons are located in lateral layers of PFC, whereas PFC-NAc are located in more medial layers within PFC. Based on this anatomical distinction, it would be interesting to determine the location of the 5 different cluster ensembles in the layers of PFC, which might suggest cells in particular clusters project to PVT or NAc. For example, are Cluster 5 neurons largely located in lateral layers of dmPFC based on their similarity in neuronal activity to PFC-PVT neurons, whereas neurons from other Clusters are located in medial layers of PFC? This analysis would provide more evidence that Cluster 5 neurons in lateral dmPFC are likely projecting to PVT and show the inhibitory activity profile, whereas neurons from different Clusters in medial dmPFC are likely projecting to Nac and show an excitatory activity profile. This analysis would also provide more concrete interpretation of the data in the Discussion section.

4. In Figure 5, D1 and D2 are denoted to indicate dmPFC activity "across days after learning (lines 621-622). Which conditioning sessions do these refer to – which session/day # relative to all sessions for each mouse? are sessions D1 and D2 the first two days in Late in Learning sessions? Are these neurons a subset of the neurons recorded during the conditioning session and if so, were they in recorded in more ventral regions of dmPFC compared to Early in Learning sessions if the lens was advanced from dorsal to ventral along dmPFC? Could there be differences in neural processing of appetitive cues in dorsal vs. ventral Cg1?

5. One major advantage of 2-photon calcium imaging is the ability to measure calcium dynamics between neurons that are recorded simultaneously (i.e. measured within the same FOV). It would be interesting perform a cross-correlation analysis to reveal if there are any significant interactions between neurons within the same FOV, determine whether they are from the same cluster, whether cells active early in learning interact with cells that are active late in learning, and whether these cross-correlations remain stable after learning.

6. Looking at activity across early vs late in learning, it was found that Cluster 1 was stable but Clusters 2-5 were not on the basis of responding to CSs. How did Clusters 2-5 change as a function of learning? Were they different based on reward delivery or licking behavior? Additional analyses are needed to strengthen this finding.

7. As it reads, the authors discuss their findings in relation to whether they agree with other studies and tools needed to answer questions about the role of specific cell types in prefrontal circuits for appetitive discrimination tasks. To strengthen the importance of this study, further discussion is needed that includes more interpretation of the data. Doing additional analyses will provide more findings to interpret, so the reader has a better grasp of the importance of this study.

8. In figure 3, CDF is undefined. Please define.

*Reviewer #3:*

In this study, Otis and colleagues evaluated the neural dynamics in the PFC governing reward learning, particularly those occurring during Pavlovian cue-sucrose associations. In particular, the study characterizes five unique neuronal ensembles exhibiting complex response patterns that seem to be relevant for the encoding of reward-predictive cues, the reward itself, and/or reward-related behavioral responses. The study also shows that the activity of these neuronal ensembles decodes behavioral variables better than chance. Interestingly, the study also shows that the activity of these neuronal ensembles during early stages of learning predicts their activity profile during late stages of learning, which remain stable afterwards.

The following list represents other strengths of the study.

1. New insights are revealed on neurophysiological mechanisms in the PFC governing cue-reward learning, using an in-vivo technique that provides great anatomical resolution, 2-photon calcium imaging.

2. This study double downs on the validity and power of head-fixed preparations to evaluate neural dynamics and their relationship to behavioral output.

3. Computational and statistical analyses are used to disentangle neuron-to-neuron variability, and to cluster neurons into distinct categories based on their response patterns.

4. In addition, throughout the study, authors describe complex methods in easy-to-understand language and illustrations to facilitate understanding of otherwise complex neurophysiological datasets and analyses.

Despite these strengths, this study could still be improved in a couple of aspects to better support the main claims and conclusions.

1. The initial analysis to cluster neurons together based on their activity patterns during the task produced somewhat confusing results in which for instance neurons exhibiting either excitatory or inhibitory responses to certain task events (e.g., either CS+, CS-, reward, or licks) were clustered together.

2. In addition, this study can greatly benefit from additional experiments (e.g., optogenetics) to manipulate neural activity during certain task epochs to test the importance of the observed activity patterns.

In general, I feel enthusiastic with the prospect of publication for this article. Though, I have several concerns that require further attention and revision to improve the overall impact of the study. As it stands, I believe this study is not yet ready for publication in *eLife*. But if my concerns are properly addressed with substantial revisions, I will feel even more enthusiastic to consider this paper for publication in *eLife*.

In the list below highlights some issues, confusions, or suggestions for additional analyses or experiments with the hopes to improve the overall quality of the study.

1. Results from the initial analysis to separate neurons into distinct neuronal ensembles are confusing. While this analysis (PCA) revealed five "unique" neuronal ensembles that supposedly encode specialized information during cue-reward learning, it is quite confusing that within-cluster responses are still very heterogeneous. For example, as shown in the Figure 1H heatmaps, within-cluster responses varied a lot across and included excitatory responses (in purple), inhibitory responses (in green), and weak responses (colors in between) within each cluster. There is also heterogeneity in the temporal profile of the responses within each cluster. How or why did neurons exhibiting excitatory and inhibitory responses get clustered together? And why did neurons exhibiting very fast responses get clustered together with neurons exhibiting slower responses? What am I missing here? Perhaps the authors could try a different clustering method (e.g., hierarchical analysis) to either confirm their clusters or potentially reveal more homogenous clusters. After all, in theory there could be many more neuronal clusters due to the many experimental variables analyzed (CS+, CS-, sucrose, sucrose omission, licks), all the possible ways neurons could respond (i.e., excitation, inhibition, no response, fast response, delayed response), and all possible response combinations (e.g., excitation to the cue, but inhibition to sucrose, etc).

2. Figure 1J summarizes the average within-cluster response patterns in PSTH form. Keeping in mind my first issue above, these PSTHs then seem misleading. For instance, the average PSTHs for Cluster-2 shows selective excitatory responses to the CS+. Yet, heterogeneity can be appreciated in the heatmaps in Figure 1H, with even some neurons exhibiting inhibitory responses to the CS-. Again, the authors should consider reinforcing or revising these results using a different clustering analysis.

3. In Figure 4, authors attempted to evaluate the evolution of response patterns in the distinct neuronal ensembles across learning. They did so by comparing ensemble activity during early versus late learning sessions. While significant Pearson correlations were detected for most ensembles during CS+ and CS-, I am not convinced that this is the best analysis to explore the evolution of neuronal activity patterns across learning. These results may just indicate that activity patterns may have developed very rapidly early in training, even before significant learning was observed at the behavioral level. To overcome this, authors could instead compare the magnitude of responses across learning sessions, or even at different segments within the early sessions (e.g., first 10 trials, versus 10 subsequent trials, and so on) to better explore whether the magnitude of responses is amplified as training progresses.

4. Caution is recommended for the type of t-test used in some analyses. For example, an independent t-test was used to compare cue discrimination scores between two days in the same subset of animals. Should this rather be a paired sample t-test?

5. Finally, all findings in this study are of correlative nature. Thus, additional experiments are needed to reinforce some of the claims raised in the study. For instance, the last sentence in the abstract says – "Our results characterize the complex dmPFC neuronal ensemble dynamics that relay learning-dependent signals for prediction of reward availability and initiation of conditioned reward seeking". If this is true, then manipulations of neural activity during certain epochs should produce significant changes in behavioral responses. A potential additional experiment could then be optogenetic-mediated inhibition during the CS+ to see whether lick rates are impaired. While this experiment could be performed in a non-ensemble-specific manner (i.e., optogenetic inhibition of all excitatory neurons in the area), it would be even better if the microscope used by the authors has holographic stimulation capabilities to selectively manipulate particular ensembles based on their response pattern. This is consistent with the last suggestion by the authors towards the end of the discussion ("functionally targeting each neuronal ensemble independently…").

[Editors' note: further revisions were suggested prior to acceptance, as described below.]

Thank you for submitting your article "Specialized coding patterns among dorsomedial prefrontal neuronal ensembles predict conditioned reward seeking" for consideration by *eLife*. Your article has been reviewed by 3 peer reviewers, and the evaluation has been overseen by a Reviewing Editor and Kate Wassum as the Senior Editor. The following individuals involved in review of your submission have agreed to reveal their identity: Maria M Diehl (Reviewer #2); Anthony Burgos-robles (Reviewer #3).

Essential Revisions:

As you will find from the evaluation summaries provided by each of the reviewers below, the general consensus is that the revised manuscript has addressed most of the critiques raised after the initial submission. However, Reviewer #1 has raised a few remaining points that we feel must be addressed for the manuscript to be considered appropriate for publication in *eLife*.

Specifically, we deem essential that in addition to authors current graphics on the relative distance of clustered neurons from the GRIN lens, they provide a quantitative analysis of how this distance influences clustering or relation to behavior. For additional details on the requested analyses please refer to the comments from Reviewer #1.

We also ask that when referring to the behavioral procedure the authors adopt terminology that most accurately represent the conditions surrounding behavioral tests (second point from Reviewer #1).

If you have not already done so, please ensure your manuscript complies with the *eLife* policies for statistical reporting: https://reviewer.elifesciences.org/author-guide/full "Report exact p-values wherever possible alongside the summary statistics and 95% confidence intervals. These should be reported for all key questions and not only when the p-value is less than 0.05."

Please include a key resource table.

*Reviewer #1:*

The authors were overall responsive to critiques and several of the issues raised in the previous reviews have been addressed. However, some concerns remain, the first of which is still requires additional analysis prior to publication.

1. The most significant remaining issue is the potential effect of differential SNR across imaging planes due to the GRIN lens properties. The fact that clusters show differential patterns and not only differences in amplitude does not negate the potential impact of SNR on the clustering – the ability to detect a differential pattern of responses between neurons is dependent on sufficient SNR, which is evidenced directly in the dataset by the fact that some of the clusters are defined by a lack of response. The authors have made great improvements on dealing with this issue from the original submission but given that essentially all the claims in the manuscript are based on the clustering analyses some quantitative assessment should be provided.

I appreciate the authors caution in using relative measurements to estimate the relative distance of clustered neurons from the GRIN lens – this is the most appropriate way to begin to approach the issue. However, the estimations are only graphically displayed, without quantitative analysis of their influence on clustering or relation to behavior, and the visualization of the data in figure 1 S5 makes it difficult to discern if there are topographical effects due to the number of overlapping points. In Figure 1 Sup 5B, how many neurons are in each line across the D/V axis? Is there a correlation between estimated location and probability of cluster membership? This is critical for determining if there is an influence of imaging plane on the clustering analysis. A complimentary approach would be to subsample and perform the clustering analysis only from a subset of DV planes at a time and determine reproducibility of cluster membership and their relationships with behavior. This is most concerning for the interpretation of Figure 2, where differential number of neurons sampled from each plane across animals could easily produce spurious correlations that reflect sampling bias rather than biological relationships.

2. Regarding the use of concurrent thirst and sucrose to motivate behavior, while it is true that head-fixed procedures often include water deprivation, these procedures were developed to motivate engagement in sensory processing tasks, not to analyze the effects of the reward itself as in the current manuscript. This is highlighted in both of the citations provided by the authors (other than their previous work) – the Goldstein et al., reference also goes on to show that how the deprivation is performed (e.g. water vs food) can dramatically impact the resulting reward-conditioned behaviors. This is not necessarily an inherent flaw in the study, but with the current wording/claims it becomes an issue.

For example, the authors refer to the behavioral procedure as 'Pavlovian sucrose conditioning' throughout – would the conditioned response (anticipatory licking) still occur if only water was delivered? Given that mice typically drink ~4 mL per day and only ~1mL is provided outside of the behavioral task, a strong argument can be made from the literature that the fluid has a much greater reinforcing/conditioning strength than the sucrose itself. I don't see any utility to empirically testing this, but given that the goal of the study is to examine conditioned reward seeking at the very least accurate terminology should be used throughout (e.g. Pavlovian conditioned licking or similar). To facilitate integration with the literature it would also be useful to add a discussion point noting that this protocol is likely to influence sucrose palatability (e.g. PMID: 16248727) as well as magnitude and nature of conditioned responses (e.g. PMID: 26913541 and 16812301).

3. The authors should clarify in the methods when the homecage water was provided in relation to behavioral testing, as well as provide an estimate of the range of total fluid and sucrose consumed in a typical session.

---

## [Author Response]

Essential Revisions:The reviewers agree that the authors' identification of dmPFC neuronal ensembles with heterogeneous coding patterns offers important insight about the neurophysiological mechanisms governing cue-reward learning. However, as independently outlined below by each one of the reviewers, there are multiple aspects of the study that must be strengthened before the paper can be considered for publication in eLife. With the exception of the optogenetic behavioral manipulations requested by Reviewer # 3, we consider that all other concerns raised by the reviewers must be addressed in full. Specifically, the authors must address:1) All technical concerns regarding the imaging technique that were raised by Reviewer # 1.

We now provide substantial new experimental data and analyses to address concerns related to imaging techniques as raised by Reviewer #1, as can be found in new figures:

Figure 1 – Supplement 2

Figure 1 – Supplement 5

Figure 1 – Video 1

Figure 4 – Supplement 1

Figure 6 – Supplement 1

Additionally, we provide clarifications throughout the methods, as described point-by-point below, to address the methodological concerns. It should be noted that we are using techniques (both imaging and behavioral) that are well characterized in the field, as we now indicate through extensive referencing.

2) All statistical and data analysis concerns raised by Reviewers #1, 2 and 3.

We have addressed all statistical and data analysis concerns. In addition to the figures listed above, we have added the following new figures to address these concerns:

Figure 1 – Supplement 1,3-5

Figure 4C, 4D

Figure 4 – Supplement 2

Figure 5

3) Additional clarifications of methods and analyses, and an improved discussion as suggested by the reviewers.

We have clarified our methods, analyses, and have broadened our discussion as described point-by-point below.

4) The concerns about the behavioral design raised by Reviewer #1.

We have addressed these concerns in the methods and discussion, which includes referencing of papers from other labs that were integral in the development of the head-fixed behavioral assay described in the manuscript. It should be noted that the same behavioral assay and associated parameters have been used extensively in the field, including by the lead PI (Otis et al., 2017 *Nature*; Otis et al., 2019 *Neuron;* Namboodiri et al., 2019 *Nature Neuroscience*), due to its power for integrating two-photon imaging with behavioral models of associative learning.

5) Due to the lack of causality, revise the text to soften the language a bit in some of the sentences describing the interpretation.

We have softened the language of the manuscript as appropriately suggested.

Reviewer #1:The manuscript from Grant et al., explores heterogeneity in coding patterns of mPFC pyramidal neurons during reward learning. The manuscript addresses a critical question in cortex and neuroscience in general – how do neuronal coding patterns lead to behavioral outputs and learned behaviors? While the manuscript takes a technically innovative approach there are multiple issues with the behavioral design, imaging, and interpretation. These issues are addressable, and the manuscript has potential for high impact in the field, but to support the current conclusions would require significant additional analysis and experimentation. Issues are listed below:1. There are major issues with the imaging methodologies, particularly with the using multiple imaging planes in each animal and with the longitudinal co-registration. Regarding the FOVs, the authors report that each imaging plane was separated by 50uM. However, since GRIN lenses display non-linear ray transformations in both the lateral and axial planes, movement of the external objective in 50uM steps cannot be assumed to produce a 50uM change in imaging plane. Indeed, in the representative images the same vasculature can be seen in multiple planes, and the veins in this area often smaller than 50uM. Though it is difficult to discern, it appears that the same cell constellations appear in multiple planes in the representatives.

The point that cells could be visualized in multiple FOVs is a valid concern, and something that we addressed rigorously prior to performing our experiments (using practices based on the lead PI’s previous studies using microendoscopic GRIN lenses or cannula for deep brain 2-photon imaging and longitudinal cell tracking; see Otis et al., 2017 Nature; Namboodiri et al., 2019 Nature Neuroscience; Otis et al., 2019 Neuron; McHenry et al., 2017 Nature Neuroscience; Rossi et al., 2019 Science). We agree that the non-linear ray transformations through microendoscopic GRIN lenses make it difficult to know exactly the distance we are traveling in the X, Y, and Z imaging planes (which we now estimate in Figure 1 – Supplement 5). This is especially true for deeper fields of view (FOVs), which will have poorer SNR and thus more cells from surrounding FOVs could unfortunately be included in the imaging field. As such, we travel at a minimum 50 microns (objective movement) between each FOV (the number generally increases with greater depth). We now describe our protocol for distinct cell visualization in more detail in the methods (page 20-21):

“Each FOV was separated by at least 50 μm of objective movement in the Z-plane to avoid visualization of the same cells in multiple FOVs. Due to non-linear ray transformation introduced by the GRIN lens, this was especially important when imaging deeper FOVs. To ensure there was no signal bleed-through from superficial FOVs, a full cell layer was visualized between each FOV. Since neurons are ~20 µm in diameter, visualization of 3 layers (two imaging planes and an inbetween plane) lead to roughly 50 µm of z-movement and isolation of unique cell layers (Figure 1 – Supplement 2).”

Additionally, we include a z-stack video (new Figure 1 – Video 1), which pauses at distinct FOVs to transparently demonstrate that the FOVs under study included non-overlapping neurons. Finally, we include a supplemental figure (new Figure 1 –Supplemental Figure 2) wherein we show three FOVs separated along the z-axis overlaid in separate colors, to better visualize the distinct cell constellations between imaging planes. Regarding the vasculature, although we agree that vessels can be small within mPFC, as seen in Figure 1 the capillaries are >3x the diameter of nearby neurons. Furthermore, the center of those capillaries is not in focus between FOVs, confirming that we are visualizing distinct imaging fields. Finally, we now clarify how distinct FOVs were identified and how ROIs were drawn to ensure identification of unique cellular layers (page 21):

“Care was taken to only assign regions of interest to visually distinct cells, and each region of interest was confirmed independently by an observer who was blind to the experimental conditions. In cases where neighboring cells or processes overlapped, regions of interest were drawn to exclude areas of overlap. The blind observer also ensured that cells were clearly in view (cells were not counted if the center-of-mass was not in focus) and that neurons were not visible in multiple FOVs (if they were, the imaging plane would not be used; n = 0). To do so, the blind observer examined z-stack videos to determine if the neurons between FOVs were independent (see Figure 1 – Video 1).”

2. Even if we are to accept that no cells were double counted, a more critical issue for the current claims of the manuscript is that collection efficiency, and therefore SNR of the recording, will be altered as a function of the distance of each neuron from the ideal focal plan of the implanted GRIN endoscope. This is a potentially critical flaw without significant additional analysis. For example, all of the clustering analysis could be highly biased by the number of neurons that were included from each imaging plane which is likely to vary from animal to animal. While this is always somewhat of a concern with GRIN imaging, because 3-4 imaging planes were used in each animal and that clustering analysis was performed on pooled data it is possible that difference in SNR across imaging planes is driving many of the effects in the manuscript.

The reviewer raises a credible concern, which we now address. First, it is worth noting SNR is unlikely to influence the clustering results shown in the manuscript, as all clusters show unique response patterns (see Figure 1G-I) rather than unique response amplitudes (only the latter would be accounted for by SNR differences. Second, we now show the estimated relative spatial location of each recorded neuron grouped by cell cluster (new Figure 1 – Supplement 5). Although the data only provide a gross estimation of relative locations that are not to scale), the figure does provide assurance that each cell cluster is represented regardless of imaging depth in our experiments (see panel B).

Due to the technical limitations of this spatial analysis (due to issues highlighted by the Reviewer), we do not make it a focus of the manuscript. Additionally, we highlight its limitations in both the methods and figure legend.

3. Regarding longitudinal registration, minimal methodological information is provided which is concerning given that this a notoriously difficult endeavor especially in dense recording such as these data. How were these data validated? Was the data set scored by a second experimenter for cross validation? What percent of neurons were tracked? Was any network analysis of cell location used to verify results?

We agree, this is an excellent point that we first address with clarification in the methods section (see page 25):

“Cell tracking was performed by a student blinded to experimental conditions to reduce the potential for experimenter-related biases. Cells were identified based on relative position and morphology across days, with conservative longitudinal tracking to prevent between-cell comparisons. Following initial cell tracking, a second experimenter confirmed accurate day-to-day cellular co-registration for cross validation.”

Second, we provide new data and new analyses to highlight the accuracy of our longitudinal coregistration across days. Specifically, we provide shuffled datasets wherein we shuffled the tracking labels for neurons within each FOV (but not between FOVs). This shuffling analysis abolished the strong correlations found when tracking neurons from early to late in learning behavioral sessions (see new Figure 4 – Supplement 1) as well as across days in late in learning (maintenance) behavioral datasets (see new Figure 6 – Supplement 1). These new data provide strong evidence that we are indeed tracking the same neurons across days in Figures 4 and 6, and that the correlated activity patterns across days are not related to general correlations in activity between neurons in the same FOVs.

4. While the issues with the imaging are critical to address, it is likely that in depth analysis could resolve the problems without the need for additional experiments. However, there is also some problems with the behavioral design – these will either require additional experiments or require that the claims of the manuscript be altered. All of the changes in mPFC dynamics that occur across the behavioral task are claimed to be related to reward learning, but there are several processes that are not parsed in the task design. For example, would some of these changes happen with habituation? Would some of these changes happen with lick rate, volume consumed? None of those are dependent on associative learning and could just as strongly predict the changes that are seen in the dataset.

We have softened the language of the text and consider alternative ideas as alluded to by the reviewer and thank them for pointing this out. Altogether, although there are several lines of evidence which posit that the measured adaptations in activity are related to associative learning (habituation is of course another type of non-associative learning), we agree that other mechanisms could be involved as described in our new discussion paragraph (page 15-16):

“One caveat to our study is the possibility that observed changes in neuronal activity could be driven by factors other than associative learning, such as nonassociative learning (e.g., habituation) or feeding in general. However, there are several lines of evidence that suggest otherwise. First, cluster-specific responses or adaptations in activity across learning were generally distinct for CS+ and CS- trials (see Figure 4), suggesting that the observed adaptations are related to cuereward associative information. Second, cue discrimination, lick, and reward decoding improved over time, despite mice consuming the reward both before and after learning (see peristimulus time histograms; Figure 1C). Finally, previous evidence in the same behavioral task reveals that both excitatory and inhibitory responses among subpopulations of dmPFC pyramidal neurons are critical for cue-reward associative learning and cue-driven licking but not licking alone (Otis et al., 2017; Otis et al., 2019). Overall, evidence suggests that identified adaptations in dmPFC activity dynamics are likely related to cue-reward associative learning. However, the possibility that there are components not related to associative learning should certainly be considered.”5. What is the justification for using water restriction combined with a sucrose solution in water as a reinforcer? Given that sucrose functions as a reinforcer without water restriction and that water functions as reinforcer under water deprived conditions, it is unclear whether the water or the sucrose is functioning as the primary reinforcer.

The justification for using water restriction along with the water + sucrose reward was to 1) facilitate appetitive learning and 2) increase the number of trials wherein the mice would perform for the reward while head restrained (through anticipatory licking). This has been characterized by Karel Svoboda’s lab as well as others (Guo et al., 2014; Goldstein et al., 2018), and has become standard practice for many of the PI’s papers within Dr. Garret Stuber’s laboratory (Otis et al., 2017; Otis et al., 2019; Namboodiri et al., 2019; Rossi et al., 2019). We now discuss the rationale for using water restriction in combination with liquid sucrose in the methods section (page 19):

“Mild water restriction facilitates appetitive learning in head-restrained mice, particularly when the reinforcer is sucrose mixed in water (Guo et al., 2014). Additionally, mild water restriction plus head restraint results in minimal signs of distress while allowing simultaneous two-photon imaging across many trials wherein a behavioral task is repeatedly performed (Guo et al., 2014; Goldstein et al., 2018; Otis et al., 2017; Otis et al., 2019; Namboodiri et al., 2019). Thus, we used water restriction in combination with Pavlovian conditioning for a liquid sucrose reward to study appetitive learning in mice…”

Reviewer #2:1. Behavioral sessions were classified as either "early in learning" or "late in learning" and are defined based on the animal's performance (using auROC) across multiple sessions of the sucrose conditioning task. The authors perform an independent t-test comparing performance in early vs. late in learning sessions; however, these 2 groups of sessions were pre-defined by the animal performance itself. Therefore, it is inappropriate to use a statistical test since the periods of early and late were selected based on the behavioral data (i.e. doing a statistical test on data that was pre-selected). A statistical test is not needed here, but the authors should emphasize that the number of early vs late in learning sessions were unique to each animal and selected based on their performance (this is only mentioned in the methods, but authors should consider reiterating this point in the results). As a reader, it was very difficult to understand how early and late in learning was defined and I had to go back and read the methods multiple times and look through the results for clarification. I initially thought early vs. late in learning referred to early trials vs. late trials within a session. If early and late was defined based on trial number instead of auROC across sessions, are the behavioral results similar to what was reported? On average, how many sessions did it take for each mouse to reach criteria?

We agree and thank the reviewer for catching that a t-test test is not appropriate considering that we used pre-determined testing criteria to include the data itself. Thus, we have removed this analysis from the results and figure/caption. Additionally, we clarify that ‘early-inlearning’ and ‘late-in-learning’ data were collected during separate behavioral sessions on separate days throughout the manuscript (both in the methods and results). Additionally, we show data for each animal illustrating learning performance across days, including the criteria for defining early and late in learning behavioral sessions (new Figure 1 – Supplement 1). Finally, we also now evaluate both behavioral and neuronal response adaptations within early-in-learning and late-in-learning behavioral sessions (across trials rather than across sessions). Overall, these new data reveal quite homogenous trial-by-trial behavioral and neuronal response patterns within sessions for each cluster (new Figure 5), rather than within-session adaptations as one might expect if learning occurred within a single session.

2. Because I was confused about early vs late in learning being within session vs. across session, I was also confused about the data analyzed in Figure 4. How many sessions were in early vs late? Was the miniscope lens advanced after each conditioning session? Was there a subset or a different set of neurons that were recorded from across multiple sessions/days? Also, why do the traces in Figure 1I look very identical to the traces in Figure 4B (lower)? Are the cells analyzed in Figure 4 a subset of those shown in Figure 1? Please clarify. I think it would also be helpful to use the same terms consistently throughout the text when referring to the conditioning sessions and clearly state at the beginning of the results that one session was recorded per day and the lens was advanced to a new FOV (if that was the case) – sometimes sessions are referred to as FOVs or different days.

Thank you for these observations, and our apologies for any confusion. For clarity, we will respond to each point individually:

How many sessions were in early vs late?

We now indicate this in the results (page 5):

“A single imaging plane (FOV) was selected during each day of training, resulting in 28 FOVs recorded early in learning (10 mice; 28 FOVs, 2092 neurons) and 21 FOVs late in learning (7 mice, 21 FOVs; 1511 neurons; 3 mice did not reach late in learning due to headcap issues)”

Was the miniscope lens advanced after each conditioning session?

We used a moveable two-photon objective to capture fields of view ranging from 0-300 µm beneath the GRIN lens. We now address this on page 20:

“Fields of view (FOVs) were visualized from 0-300 µm beneath the GRIN lens and were selected before “early” behavioral imaging sessions. Each FOV was separated by at least 50 μm of objective movement in the Z-plane to avoid visualization of the same cells in multiple FOVs”

Was the miniscope lens advanced after each conditioning session? Was there a subset or a different set of neurons that were recorded from across multiple sessions/days?

Yes, we were able to visually track only a subset of neurons across learning, from early to late in learning behavioral sessions. Thus, Figure 1l and 4a are identical because they are the subset of neurons that were tracked across learning, as the Reviewer has indicated. We apologize for the lack of clarity on this issue, and now provide further clarification throughout the methods and results (e.g., page 10):

“Two-photon microscopy enables visual tracking of single, virally labeled neurons across days (Namboodiri et al., 2019; Otis et al., 2017). Thus, we were able to track a subset of the above dmPFC excitatory output neurons from early to late in learning behavioral sessions (n = 5 mice; 9 FOVs; 416 neurons) to evaluate neuronal response evolution across learning”.

I think it would also be helpful to use the same terms consistently throughout the text when referring to the conditioning sessions and clearly state at the beginning of the results that one session was recorded per day and the lens was advanced to a new FOV (if that was the case) – sometimes sessions are referred to as FOVs or different days.

We apologize for the general lack of consistency and agree that this requires adjustments. We have therefore clarified throughout the manuscript as suggested such that we refer to recordings as early in learning or late in learning. Furthermore, we provide a figure to show that early and late in learning sessions are on different days of training (Figure 1 – Supplement 1). Finally, we now clearly state in the results that one session was recorded per day. For example, on page 20:

“Mice readily acquired this task across sessions (one session per day; see Figure 1 – Supplement 1), showing conditioned licking behavior between the CS+ offset and reward delivery (trace interval), but not the CS- offset and equivalent no reward epoch during sessions on later days (deemed ‘late in learning; Figure 1CD).”

3. PFC-PVT neurons are located in lateral layers of PFC, whereas PFC-NAc are located in more medial layers within PFC. Based on this anatomical distinction, it would be interesting to determine the location of the 5 different cluster ensembles in the layers of PFC, which might suggest cells in particular clusters project to PVT or NAc. For example, are Cluster 5 neurons largely located in lateral layers of dmPFC based on their similarity in neuronal activity to PFC-PVT neurons, whereas neurons from other Clusters are located in medial layers of PFC? This analysis would provide more evidence that Cluster 5 neurons in lateral dmPFC are likely projecting to PVT and show the inhibitory activity profile, whereas neurons from different Clusters in medial dmPFC are likely projecting to Nac and show an excitatory activity profile. This analysis would also provide more concrete interpretation of the data in the Discussion section.

We agree, identification of the anatomical location of each neuron (and therefore cluster) within dmPFC would provide insight into layer-specific activity dynamics, including those that may control behavior. Thus, we have done our best to approximate the relative position of each neuron, grouped by cluster, within dmPFC (see Figure 1 – Supplement 5). These data show the vast heterogeneity in dmPFC, such that each cluster is represented across axes in our recordings.

It should be noted that there are major caveats to this cell mapping analysis due to methodological challenges that we could not overcome, significant enough that we discuss them up front in the methods and in the supplementary figure legend. Additionally, we do not focus on this data in the main body of the manuscript, and only briefly refer to the data in the Results section. Broadly speaking, it is extremely difficult (if not impossible, at least for us) to determine if neurons were in specific layer of dmPFC due to minor deviations in GRIN lens angle, placement (which are difficult to identify post-mortem as the tissue surrounding the lens is generally damaged when pulling out the lens), headcap orientation (a tilted headcap changes how the light focuses beneath the lens), and due to non-linear ray trasformations through the lens itself (brought up by Reviewer #1). Additionally, because dmPFC is not as well layered as other cortices, it is very challenging to see the layers in the first place. Thus, we were hesitant to include these data in the manuscript, as they could be interpreted as defined locations rather than extremely gross estimates of relative locations. We would like to hear more from the Reviewer(s) and Editor as to whether they think this data is worth including as is, or if it could rather be misleading.

4. In Figure 5, D1 and D2 are denoted to indicate dmPFC activity "across days after learning (lines 621-622). Which conditioning sessions do these refer to – which session/day # relative to all sessions for each mouse? are sessions D1 and D2 the first two days in Late in Learning sessions? Are these neurons a subset of the neurons recorded during the conditioning session and if so, were they in recorded in more ventral regions of dmPFC compared to Early in Learning sessions if the lens was advanced from dorsal to ventral along dmPFC? Could there be differences in neural processing of appetitive cues in dorsal vs. ventral Cg1?

The data shown in Figure 6 refer to two conditioning sessions, each recorded late in learning. The first (session D1) shows data from the first day the specified FOV was eligible to be recorded late in learning (based on the animal’s cue discrimination). The second (session D2) shows data from a second late in learning session, in which the same FOV was recorded from 48 hours or more after session D1. There is no lens movement as these are two-photon recordings not miniscope recordings. We have clarified this in both the methods and Results sections and thank the reviewer for pointing out that this was previously unclear (e.g., see page 25):

“Specific neurons could be reliably identified across days based on structure and relative position within each FOV (Figure 4A), allowing us to evaluate the evolution and maintenance of activity in single neurons across days. To do so, single cell tracking was performed from early to late in learning behavioral sessions to determine neuronal response evolution across learning (Figure 4). Additionally, single cell tracking was performed across two late in learning behavioral sessions each separated by a minimum of 48 hours to evaluate post-learning response adaptation or maintenance (Figure 6).”

5. One major advantage of 2-photon calcium imaging is the ability to measure calcium dynamics between neurons that are recorded simultaneously (i.e. measured within the same FOV). It would be interesting perform a cross-correlation analysis to reveal if there are any significant interactions between neurons within the same FOV, determine whether they are from the same cluster, whether cells active early in learning interact with cells that are active late in learning, and whether these cross-correlations remain stable after learning.

As suggested by the Reviewer, we have performed extensive cross correlation analyses on all our neurons early and late in learning, grouped by cluster. We show data from tracked neurons so that we could split neurons by cluster both early in learning and late in learning. Interestingly, neurons within the same cluster have little to no lag both early in learning and late in learning, whereas neurons in different clusters generally do have lag (see new Figure 4 – Supplement 2). These data suggest that there may be within-cluster similarities in activity even before appetitive learning has occurred.

While we agree with the Reviewer that this is an interesting analysis, one issue that limits its usefulness is the relatively slow dynamics of the calcium sensor and the speed of imaging itself. This prevents identification of spike-to-spike correlations in activity across neurons, and rather only allows a somewhat slow approximation of correlated activity patterns (which is a far cry from suggesting causal relationships in activity). Thus, while we believe the cross-correlation analysis that we have performed does add to the manuscript, we have not made it a primary focus due to this significant limitation which hinders data interpretation.

6. Looking at activity across early vs late in learning, it was found that Cluster 1 was stable but Clusters 2-5 were not on the basis of responding to CSs. How did Clusters 2-5 change as a function of learning? Were they different based on reward delivery or licking behavior? Additional analyses are needed to strengthen this finding.

Based on these questions we have added additional data and analysis to examine mean response evolution during CS+ and CS- trials (new Figure 4C-D). As can be seen in Figure 4, the only cluster that responds early in learning and does not adapt across learning is indeed Cluster 1. Differential encoding of these newly formed clusters is already a primary focus of Figure 3, wherein we show differential encoding patterns of each cluster for the cues, reward, and licking after learning. Thus, these newly developed responses are specific to the cues, reward, and licking as shown in Figure 3. Our reasoning for not redoing the decoding on tracked datasets is that such analysis requires extremely high power (generally >250 neurons) and given the low number of neurons that were difficult to track (currently 417 split into 5 clusters), we cannot sufficiently power the analysis for tracking datasets in Figure 4 (and find that it is unnecessary based on findings in Figure 3).

7. As it reads, the authors discuss their findings in relation to whether they agree with other studies and tools needed to answer questions about the role of specific cell types in prefrontal circuits for appetitive discrimination tasks. To strengthen the importance of this study, further discussion is needed that includes more interpretation of the data. Doing additional analyses will provide more findings to interpret, so the reader has a better grasp of the importance of this study.

In response to the excellent suggestions by the Reviewers, we have performed new analyses and show new data ( = new Figure 1 – Video 1; Figure 1 – Supplements 1, 2, 3, 4 and 5; Figure 4C-D; Figure 4 – Supplements 1 and 2; Figure 5; and Figure 6 – Supplement 1) to provide more findings for interpretation. We refer to these new data and analyses throughout the manuscript (see all new changes in red) and have lengthened the discussion considerably to adequately interpret and address the impact of these new findings. We present what we hope is a much-improved manuscript thanks to the reviewers.

8. In figure 3, CDF is undefined. Please define.

Thank you to the reviewer for catching this error, we now define CDF in the legend.

Reviewer #3:In the list below highlights some issues, confusions, or suggestions for additional analyses or experiments with the hopes to improve the overall quality of the study.1. Results from the initial analysis to separate neurons into distinct neuronal ensembles are confusing. While this analysis (PCA) revealed five "unique" neuronal ensembles that supposedly encode specialized information during cue-reward learning, it is quite confusing that within-cluster responses are still very heterogeneous. For example, as shown in the Figure 1H heatmaps, within-cluster responses varied a lot across and included excitatory responses (in purple), inhibitory responses (in green), and weak responses (colors in between) within each cluster. There is also heterogeneity in the temporal profile of the responses within each cluster. How or why did neurons exhibiting excitatory and inhibitory responses get clustered together? And why did neurons exhibiting very fast responses get clustered together with neurons exhibiting slower responses? What am I missing here? Perhaps the authors could try a different clustering method (e.g., hierarchical analysis) to either confirm their clusters or potentially reveal more homogenous clusters. After all, in theory there could be many more neuronal clusters due to the many experimental variables analyzed (CS+, CS-, sucrose, sucrose omission, licks), all the possible ways neurons could respond (i.e., excitation, inhibition, no response, fast response, delayed response), and all possible response combinations (e.g., excitation to the cue, but inhibition to sucrose, etc).

This is an understandable concern that we address through the addition of new data, analysis, and a more thorough description of the data. First, we used PCA to reduce the dimensionality of our dataset, followed by spectral clustering to identify unique activity patterns among neurons (i.e., “neuronal ensembles”). The PCs used to inform the clustering algorithm can now be found in new Figure 1 – Supplement 3A-B. This methodology is based on previous papers showing the suitability of spectral clustering for evaluating heterogenous and dynamic response patterns in cortical circuits (e.g., Namboodiri et al., 2019). The results from this PCA analysis hopefully provide a better framework for the readers to understand the dynamic principal components that were used for the clustering analyses, which is likely the reason that not all cells within each cluster look (by eye) like they obviously fit.

In the initial submission of our manuscript, we did not discuss the alternative clustering approaches that we took to analyze our data, which specifically included agglomerative (“hierarchical”) and k-means clustering. We now add these analyses and the resulting data to the manuscript, as was insightfully suggested by the Reviewer. Overall, our preliminary analysis showed that spectral clustering outperformed agglomerative and k-means algorithms, as evidenced by improved “fitting” of each neuron within its corresponding cluster (see example silhouette plots in new Figure 1 – Supplement 3C-E). Additionally, we show how each algorithm caused separation of neurons into clusters (see new Figure 1 – Supplement 4 for agglomerative and k-means clustering; Figure 1G-I shows spectral clustering results that were included in the initial submission). The idea that spectral clustering results in improved fitting of each neuron into its corresponding cell cluster is observable by looking at the resulting clusters in Figure 1 – Supplement 4. Specifically, although the clusters overall look similar across algorithms, agglomerative and k-means clustering both result in large groups of neurons in Cluster 2 that do not seem to fit well, as compared the spectral clustering dataset (as is shown in Figure 1G-I).

Although spectral clustering seems to be the best tool to separate dmPFC neuronal responses in our task into distinct groups, the Reviewer brings up a critical point that the clustering method is not perfect. Simply stated, we are trying to simplify a heterogenous pattern of activity into homogeneous response profiles, which although is possible to some degree does is unlikely to be possible in total. We now describe this in the discussion, as we feel it is an important point to consider not only for our dataset but for most neuronal clustering datasets (see page 15):

“An important consideration for our study is the existence of heterogeneity not only between experimenter-defined cell clusters, but also within these clusters. We chose to use spectral clustering to define unique response dynamics in dmPFC neurons, as opposed to other methods (e.g., agglomerative, k-means), based on preliminary analyses (see Figure 1 – Supplement 3 and 4). Although these analyses suggested that spectral clustering provides the best separation of dmPFC neurons into groups, likely due to its ability to separate dynamic and non-singular response features, these methods overall are far from perfect. Specifically, we are trying to simplify a heterogeneous pattern of activity into homogeneous groups, which given current methodologies is only possible to some degree. Further advancement of clustering and other computational methodologies should continue to improve our ability to detect and understand unique response patterns within complex brain circuits.”

2. Figure 1J summarizes the average within-cluster response patterns in PSTH form. Keeping in mind my first issue above, these PSTHs then seem misleading. For instance, the average PSTHs for Cluster-2 shows selective excitatory responses to the CS+. Yet, heterogeneity can be appreciated in the heatmaps in Figure 1H, with even some neurons exhibiting inhibitory responses to the CS-. Again, the authors should consider reinforcing or revising these results using a different clustering analysis.

This is an understandable point, but as discussed above the heterogeneity observed in each cluster is a result of how the clustering algorithm handles the non-singular response dynamics found in our neuronal dataset (see PCs in Figure 1 – Supplement 3), and its inability to separate the neurons into perfectly unique clusters. Alternative clustering methods that we have used have not resulted in improved clustering, but rather poorer performance (see Figure 1 – Supplement 3 and 4). Furthermore, the clusters identified here are encoding unique task features (Figure 3), confirming that the identified groups of neurons may be uniquely relevant for behavioral control and are separating appropriately.

Due to this within-cluster heterogeneity, we feel that showing the responses of all neurons for each cluster (in the peristimulus heat maps) and the average response across neurons for each cluster (in the peristimulus line plots) is critical for data transparency and description. Thus, we have chosen to include both within the main figure.

3. In Figure 4, authors attempted to evaluate the evolution of response patterns in the distinct neuronal ensembles across learning. They did so by comparing ensemble activity during early versus late learning sessions. While significant Pearson correlations were detected for most ensembles during CS+ and CS-, I am not convinced that this is the best analysis to explore the evolution of neuronal activity patterns across learning. These results may just indicate that activity patterns may have developed very rapidly early in training, even before significant learning was observed at the behavioral level. To overcome this, authors could instead compare the magnitude of responses across learning sessions, or even at different segments within the early sessions (e.g., first 10 trials, versus 10 subsequent trials, and so on) to better explore whether the magnitude of responses is amplified as training progresses.

We agree, this is an excellent point that we now address through the addition of new data.

First, as suggested by the Reviewer, we compare the magnitude of responses of responses across learning sessions (see new Figure 4C-D). Overall, the results demonstrate that the magnitude of responses during CS+ and/or CS- trials changed for Clusters 2-5, but not Cluster 1. These results confirm neuronal response adaptation for Clusters 2-5, but not Cluster 1, across learning.

Second, as also insightfully suggested by the Reviewer we compare the magnitude of behavioral and neuronal responses (separated by cluster) within sessions (first 10 trials versus last 10 trials; see new Figure 5). Results are almost identical from trials at the beginning of each session versus trials at the end of each session, suggesting that behavioral and neuronal responses may not rapidly adapt within sessions (in the task used here).

4. Caution is recommended for the type of t-test used in some analyses. For example, an independent t-test was used to compare cue discrimination scores between two days in the same subset of animals. Should this rather be a paired sample t-test?

This is a good point, and something similarly brought up by Reviewer #2. We have therefore removed the inappropriate independent t-test (Figure 1D; see response to Reviewer #2, Point #1) or changed the t-test to paired rather than independent as suggested here (Figure 6A).

5. Finally, all findings in this study are of correlative nature. Thus, additional experiments are needed to reinforce some of the claims raised in the study. For instance, the last sentence in the abstract says – "Our results characterize the complex dmPFC neuronal ensemble dynamics that relay learning-dependent signals for prediction of reward availability and initiation of conditioned reward seeking". If this is true, then manipulations of neural activity during certain epochs should produce significant changes in behavioral responses. A potential additional experiment could then be optogenetic-mediated inhibition during the CS+ to see whether lick rates are impaired. While this experiment could be performed in a non-ensemble-specific manner (i.e., optogenetic inhibition of all excitatory neurons in the area), it would be even better if the microscope used by the authors has holographic stimulation capabilities to selectively manipulate particular ensembles based on their response pattern. This is consistent with the last suggestion by the authors towards the end of the discussion ("functionally targeting each neuronal ensemble independently…").

Yes, we completely agree with the assessment of the Reviewer that this study is missing some sort of functional manipulation. However, we believe that the ensemble-specific nature of these manipulations is critical, since non-ensemble-specific manipulation could produce equal and opposing effects on behavior (Otis et al., 2017) and would poorly inform us of how distinct ensemble dynamics contribute to behavior. Despite the necessity of these experiments, current viral strategies for simultaneous recording and manipulation of neural activity are limited and have provided us with many technical challenges (though we have tried extensively based on viral approaches used by Rafael Yuste, Karel Svoboda, Karl Deisseroth, and others). As these technologies improve, we hope to be among the first groups to perform ensemble-specific manipulations in dmPFC through a GRIN lens during reward seeking. Given these technological challenges, to address the reviewer’s comment, we have softened the language regarding the impact of our findings and stress the need for improved neurotechnologies to undertake these experiments (e.g., see page 16).

[Editors' note: further revisions were suggested prior to acceptance, as described below.]

Essential Revisions:Reviewer #1:The authors were overall responsive to critiques and several of the issues raised in the previous reviews have been addressed. However, some concerns remain, the first of which is still requires additional analysis prior to publication.1. The most significant remaining issue is the potential effect of differential SNR across imaging planes due to the GRIN lens properties. The fact that clusters show differential patterns and not only differences in amplitude does not negate the potential impact of SNR on the clustering – the ability to detect a differential pattern of responses between neurons is dependent on sufficient SNR, which is evidenced directly in the dataset by the fact that some of the clusters are defined by a lack of response. The authors have made great improvements on dealing with this issue from the original submission but given that essentially all the claims in the manuscript are based on the clustering analyses some quantitative assessment should be provided.I appreciate the authors caution in using relative measurements to estimate the relative distance of clustered neurons from the GRIN lens – this is the most appropriate way to begin to approach the issue. However, the estimations are only graphically displayed, without quantitative analysis of their influence on clustering or relation to behavior, and the visualization of the data in figure 1 S5 makes it difficult to discern if there are topographical effects due to the number of overlapping points. In Figure 1 Sup 5B, how many neurons are in each line across the D/V axis? Is there a correlation between estimated location and probability of cluster membership? This is critical for determining if there is an influence of imaging plane on the clustering analysis. A complimentary approach would be to subsample and perform the clustering analysis only from a subset of DV planes at a time and determine reproducibility of cluster membership and their relationships with behavior. This is most concerning for the interpretation of Figure 2, where differential number of neurons sampled from each plane across animals could easily produce spurious correlations that reflect sampling bias rather than biological relationships.

We have now quantified the number of neurons along the A/P, M/L, and D/V axis relative to the number of total neurons for each cluster. These data are displayed as histograms along each axis in Figure 1- S5 and provide further confirmation that there is a spread of each identified cluster along each axis. Although we appreciate the Reviewer’s concern that cells could be difficult to identify if not in superficial layers, this simply is not the case as can be visualized in Figure 1 – S5. Neurons in all clusters are in the middle and deeper layers.

2. Regarding the use of concurrent thirst and sucrose to motivate behavior, while it is true that head-fixed procedures often include water deprivation, these procedures were developed to motivate engagement in sensory processing tasks, not to analyze the effects of the reward itself as in the current manuscript. This is highlighted in both of the citations provided by the authors (other than their previous work) – the Goldstein et al., reference also goes on to show that how the deprivation is performed (e.g. water vs food) can dramatically impact the resulting reward-conditioned behaviors. This is not necessarily an inherent flaw in the study, but with the current wording/claims it becomes an issue.For example, the authors refer to the behavioral procedure as 'Pavlovian sucrose conditioning' throughout – would the conditioned response (anticipatory licking) still occur if only water was delivered? Given that mice typically drink ~4 mL per day and only ~1mL is provided outside of the behavioral task, a strong argument can be made from the literature that the fluid has a much greater reinforcing/conditioning strength than the sucrose itself. I don't see any utility to empirically testing this, but given that the goal of the study is to examine conditioned reward seeking at the very least accurate terminology should be used throughout (e.g. Pavlovian conditioned licking or similar). To facilitate integration with the literature it would also be useful to add a discussion point noting that this protocol is likely to influence sucrose palatability (e.g. PMID: 16248727) as well as magnitude and nature of conditioned responses (e.g. PMID: 26913541 and 16812301).

We agree and thank the reviewer for making this distinction. We have now modified the text where applicable, for example to read “Pavlovian conditioned licking” as opposed to “Pavlovian sucrose conditioning” which seems more appropriate (for example, see page 4):

“Here we use in vivo two-photon calcium imaging to measure and longitudinally track the activity dynamics of single dmPFC excitatory output neurons throughout a Pavlovian conditioned licking task.”

In addition, we have added a discussion point as the reviewer rightfully suggested to address the issue of water restriction and palatability potentially influencing our results (see page 16):

“Another consideration to note is that the observed conditioned licking responses could be influenced by the effects of water restriction, as well as the ratio of sucrose/water used as a reward in our behavioral paradigm (Davey and Cleland, 1982; Harris and Thein, 2005; Tabbara et al., 2016). However, we did not test the influence of these variables in the current experiments.

3. The authors should clarify in the methods when the homecage water was provided in relation to behavioral testing, as well as provide an estimate of the range of total fluid and sucrose consumed in a typical session.

This information has been added to the Methods section (see page 19, 20):

“(12.5% sucrose in water, ~2.0 μl per droplet, ~0.1 mL total per session)”

“Mice received ~1 mL of water placed in a dish in their home cage after each conditioning session.”